# Discovering highly potent antimicrobial peptides with deep generative model HydrAMP

Paulina Szymczak [1,5], Marcin Możejko [1,5], Tomasz Grzegorzek [1,2], Radosław Jurczak [1], Marta Bauer [3], Damian Neubauer [3], Karol Sikora [3], Michał Michalski [4], Jacek Sroka[1], Piotr Setny [4], Wojciech Kamysz [3] & Ewa Szczurek [1] ✉

Antimicrobial peptides emerge as compounds that can alleviate the global health hazard of antimicrobial resistance, prompting a need for novel computational approaches to peptide generation. Here, we propose HydrAMP, a conditional variational autoencoder that learns lower-dimensional, continuous representation of peptides and captures their antimicrobial properties. The model disentangles the learnt representation of a peptide from its antimicrobial conditions and leverages parameter-controlled creativity. HydrAMP is the first model that is directly optimized for diverse tasks, including unconstrained and analogue generation and outperforms other approaches in these tasks. An additional preselection procedure based on ranking of generated peptides and molecular dynamics simulations increases experimental validation rate. Wet-lab experiments on five bacterial strains confirm high activity of nine peptides generated as analogues of clinically relevant prototypes, as well as six analogues of an inactive peptide. HydrAMP enables generation of diverse and potent peptides, making a step towards resolving the antimicrobial resistance crisis.

Microbes pose a continuously increasing threat to human health, in particular by causing sepsis, post-surgical infections, and putting at risk patients with chronic conditions or immunodeficiency[1]. It is estimated that microbial infection will become the main cause of death by 2050, exceeding the currently dominating cancer and cardiovascular diseases[2]. The reason for the growing danger of bacteria is their ability to gain resistance to antibiotics[1]. In the recent years, antimicrobial peptides (AMPs) are investigated as attractive alternatives to conventional antibiotic treatment. The acquisition of resistance to AMPs in bacteria is much slower[3], moreover, they can be active against pathogens that are resistant to antibiotics[4].

Typically, AMPs are amphiphatic−cationic amino acids build the hydrophylic face of the peptide, while hydrophobic residues dominate the opposite side of the molecule. Amphiphacity together with high charge allow AMPs to invade and disrupt the negatively charged bacterial cellular membrane[5]. Antibacterial activity of a peptide is measured experimentally by determining its Minimal Inhibitory Concentration (MIC). The most prominent peptides have low values of MIC, meaning that they remain effective even in low concentrations, but their prevalence is limited. Given the high therapeutic promise of AMPs, it is critical to design novel peptides that are nonexistent in nature and could be synthesised and used to treat microbial infections in the clinic.

[1]Faculty of Mathematics, Informatics and Mechanics, University of Warsaw, Stefana Banacha 2, 02-097 Warsaw, Poland. [2]NVIDIA, 2788 San Tomas Expressway, Santa Clara, CA 95051, USA. [3]Department of Inorganic Chemistry, Faculty of Pharmacy, Medical University of Gdańsk, Al. Gen. J. Hallera 107, 80-416 Gdańsk, Poland. [4]The Centre of New Technologies, University of Warsaw, Stefana Banacha 2c, 02-097 Warsaw, Poland. [5]These authors contributed equally: Paulina Szymczak, Marcin Możejko ✉e-mail: szczurek@mimuw.edu.pl

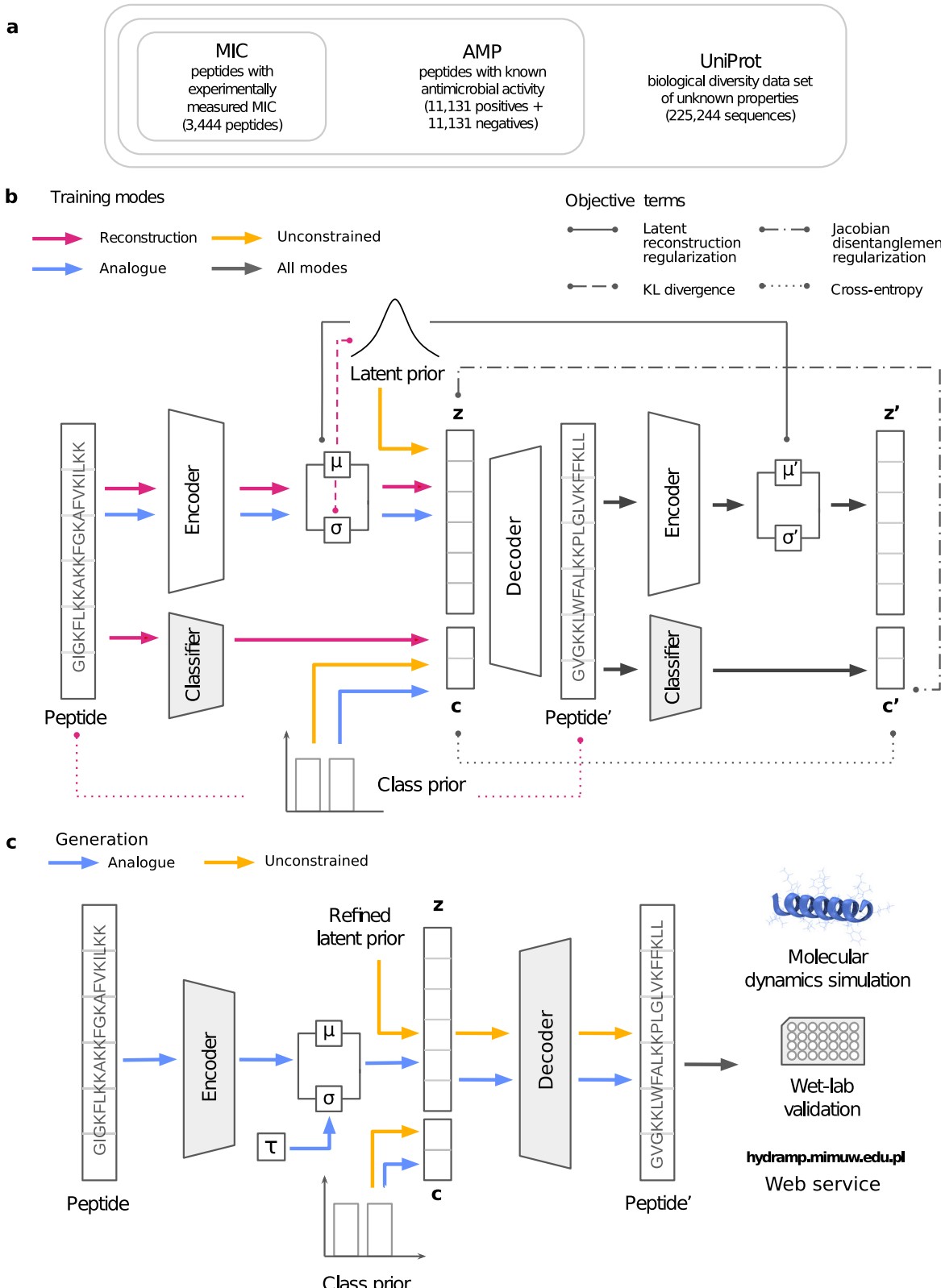

**Fig. 1 | HydrAMP architecture and data traversion overview. a** Compositional structure of the training data set. **b** Data flow and optimization setup during the training. Colors indicate training modes: reconstruction (pink), unconstrained (yellow), analogue (blue), and all modes together (grey). The arrows show the path each peptide traverses within a given training mode. Lines connect arguments of different terms of the objective (loss) function, as indicated by line styling: latent reconstruction regularization, Jacobian disentanglement regularization, KL divergence, and cross entropy. Shaded areas indicate components with frozen weights. **c** Data flow during the generation of peptides. Colors indicate generation: analogue (blue), and unconstrained (yellow). The model is validated using molecular dynamics simulations and wet-lab validation (activity and toxicity assays). HydrAMP functionality is available via a web service https://hydramp.mimuw.edu.pl/.

In biological labs, the process of identifying new antimicrobial peptides commonly proceeds by taking existing, known AMPs as prototypes, and adding or substituting amino acids, usually aiming at increasing the resulting amphiphacity and/or charge. Such generated peptides are subjected to synthesis attempt, and if synthesizable, their antimicrobial activity is experimentally verified. First, this process is tedious, and time and cost-consuming. Second, it is difficult to improve existing AMPs, which already have good physicochemical properties. Finally, even if new candidates are obtained in this way, the novel peptides will be similar in their sequence to existing peptides, and as such their diversity is expected to be poor. Thus, there is a need to devise efficient and accurate in silico approaches to novel AMPs generation.

The problem of modelling AMPs was undertaken by a number of different computational approaches. One group of these approaches are classifiers, which take a peptide as input and their task is to predict whether the peptide is an AMP or not[6–9], whether it is toxic[10,11], or whether it is active[12,13]. A related group of methods are quantitative structure-activity relationship (QSAR) models[14–16]. QSAR approaches identify a set of structural features for a given peptide, all of which are associated with the peptides being AMP, for example, helical structure or amphipathicity. Next, the model is applied to a peptide database and peptides with the highest scores for features associated with being an AMP are chosen. As such, the QSAR methods can only score existing peptides and are not able to directly generate new ones. Another approach to peptide generation is to use autoregressive models trained on AMP sequences[17–19]. To generate new peptides, these models operate in an iterative manner. In each iteration, a subsequence of the peptide constructed so far is given at input, and the model is used to propose the consecutive amino acid in the sequence. Other approaches are based on the genetic algorithms[12,20–22]. These methods iteratively evolve a population of peptide sequences by adding random mutations, evaluating their fitness, and performing cross-over and other evolutionary operations. Their performance depends on the choice of the method for introducing mutations and for evaluating their fitness. Finally, there are linguistic models[23], that consider peptides as a formal language with grammar and vocabulary. By inserting alpha-helical patterns into AMP sequences in a sliding window manner, they are able to generate novel sequences in few attempts.

Working in the peptide space (both amino acid sequence, as well as atomic composition, e.g. encoded with graph-based representations such as the Simplified Molecular Input Line Entry Specification, or shortly SMILES[24]), as it is the case with the QSAR, genetic algorithm-based methods, and linguistic approaches, has serious disadvantages. First, the sequence space is sparse. Second, it is combinatorial and discrete, and thus highly dimensional, causing the approaches working in sequence space to be computationally demanding and prone to quickly getting stuck in local minima[22]. On top of that, similarity in this space does not imply similarity of peptide function. Specifically, it is likely that operations such as amino acid substitutions, deletions, or additions, making small changes to the sequence, have large impact on amphiphacity or charge, affecting the antimicrobial activity of the peptide as well. Thus, it is desirable for the computational approach to find a better, continuous, and dimensionality-reduced representation of peptides, and operate on such representations instead.

These types of representation learning approaches to peptide generation include generative adversarial networks (GANs)[25,26] and variational autoencoders (VAEs)[13,27] as well as their conditional variants cGANs[28] and cVAEs[29,30]. The conditional variants enable generation of peptides satisfying a given condition. In contrast to VAE, training of GANs was reported to face substantial technical obstacles, such as training instabilities and mode collapse[28].

On top of that, the existing approaches are not explicitly trained to perform all the desired tasks. Specifically, almost all of the above models, except for the genetic algorithm, linguistic models and cVAEs, are suitable only for generation of peptides de novo, to which we refer as *unconstrained generation*. In fact, generation starting off from a prototype sequence, which we call the *analogue generation*, should also be optimized during training. Moreover, ideally, in the analogue generation, the peptides should be generable both from known active AMPs (positives) and non-AMPs (negatives). Indeed, the former mode would allow to directly mimic the experimental approach, while the latter is expected to increase the diversity of the pool of generated peptides. To our knowledge, none of the existing approaches are trained directly to improve non-AMPs. Finally, the results of the generative models are rarely experimentally verified, the code of the methods often is not publicly accessible, and their functionality usually is not made available to non-technical users in a digestible manner, e.g. as a web service. Specifically, from 13 previously published tools that we evaluated[13,17,20,23,27–35] five did not experimentally validate their results. The reproducibility of these 13 methods was largely limited, as nine made the code available, nine made the training data available, four shared the generated peptides, and only two methods shared all these three resources at once (Supplementary Data 1). Finally, none of these methods provided an accessible web service.

Here, we propose HydrAMP, a novel approach to peptide generation, designed to address these needs. HydrAMP is a cVAE-based model, which is specifically trained to perform analogue generation both from positives and negatives, as well as unconstrained generation. It learns a hidden space of meaningful peptide representations, which is disentangled from the set of antimicrobial conditions that a generated peptide is expected to satisfy − whether it is supposed to be a highly active AMP or not. The model is available as a web-service at https://hydramp.mimuw.edu.pl. We introduce an additional preselection procedure based on external AMP classifiers and molecular dynamics (MD) simulations. We validate the activity of generated peptides against five bacterial strains, including both gram-positive and negative, as well as antibiotic-resistant ones. We additionally measure the toxicity to mammalian erythrocytes. Given the performance proven in both computational and experimental evaluation, HydrAMP is a step forward in the daunting task of generating novel, highly active AMPs and fighting the problem of antimicrobial resistance.

## Results

### HydrAMP − a conditional, generative model of peptide sequences

HydrAMP is a model for generation of novel peptide sequences satisfying given antimicrobial activity conditions. A pair of conditions, denoted $\mathbf{c} = (c_{AMP}, c_{MIC})$ specifies whether the generated peptide is supposed to be antimicrobial (condition $c_{AMP}$) and whether it is supposed to have high antimicrobial activity, or, equivalently, low MIC (condition $c_{MIC}$). Despite the fact that the feature of being AMP and being highly active are strongly related, we keep them as separate conditions, because of the existence of peptides that are known to be antimicrobial but have low activity (see Methods).

The training data for HydrAMP consists of a curated data set of peptide sequences of up to 25 amino acids, including sequences that are known to be AMP, sequences for which MIC measurements are reported, and non-redundant sequences collected from UniProt (see Fig. 1a and Section Methods for details). The MIC value for a peptide depends on the peptide's sequence and bacterial strain. To obtain a single MIC measurement for each peptide in the training set, we followed Witten & Witten[12] and picked *Escherichia coli* (*E. coli*) as the species with the most abundant MIC measurements and averaged the MIC values per peptide across strains (see Section Methods for details). The model is trained in three modes: *reconstruction*, *analogue* and *unconstrained* (Fig. 1b). Training in the reconstruction mode facilitates the model to properly capture peptide sequences

distribution, as well as those properties that make them antimicrobial and active. This is achieved by ensuring that the reconstructed peptides are similar to the input peptides from the training data and satisfy the same conditions. In the analogue generation mode, the model is trained to generate analogues based on the provided prototype peptide and satisfying a specified condition. Finally, during the unconstrained generation mode, the model is trained to generate peptides de novo that resemble training data and satisfy the specified condition.

More formally, HydrAMP is an extension of a conditional variational autoencoder (cVAE)[36]. The model is optimized to create a meaningful, latent, real-valued vector space representation of peptides, which is easier to sample from and has a lower dimension than the original, highly dimensional and combinatorial space of peptide sequences. Apart from standard neural network-based sub-models such as *Encoder* and *Decoder*, used in the cVAE framework to operate on the latent representation in a probabilistic manner, the model also utilizes a *Classifier*. The Classifier is a neural network itself, which, unlike the Encoder and the Decoder, is pre-trained prior to HydrAMP training, and is used to classify whether an given peptide is an AMP or not, and whether it has a low MIC against *E. coli* or not. The performance of the Classifier is shown in Supplementary Table 1.

HydrAMP utilizes a number of regularization terms: latent reconstruction regularization, KL divergence, and Jacobian disentanglement regularization (Fig. 1b). The former two regularization terms are standard in the cVAE framework. The latter is specifically introduced in this work for obtaining a disentanglement between the latent representation of peptides and the condition. In this way, the latent space encodes the property of being a peptide, while the condition independently encodes whether this peptide is supposed to be an AMP or have low MIC against *E. coli*. See Methods for a detailed formal description of the model.

HydrAMP offers two types of generation: *analogue* and *unconstrained* (Fig. 1c). The analogue generation improves upon the common practice of novel peptide discovery followed in experimental labs. In contrast to the tedious trial and error process of changing the original sequence, with the prototype and the desired condition specified the model manipulates the latent representation of the prototype instead. Given the favorable nature of the latent space, which spans real-valued vectors that were trained as representations of valid peptides, and given that distances in the latent space should reflect the dissimilarities between the peptides, the points in the latent space in close proximity to the point representing the original peptide are good candidates for analogue samples. For such samples from the latent space, the role of the Decoder is then to generate a sequence of amino acids satisfying the desired condition. In this respect, HydrAMP benefits from an additional temperature parameter $\tau$ that controls the creativity of analogue generation (see Generation modes of peptides for more details). Intuitively, the temperature influences the radius with respect to the prototype point in the latent space within which the analogues are searched for, and by this, their similarity to the prototype. Compared to the analogue generation, the unconstrained generation is more standard for cVAE and enables generation of valid peptides with either random, or desired condition. To this end, samples are generated from a prior distribution over the latent space, and Decoder is used to produce sequences with the fixed condition.

## HydrAMP outperforms other models in the ability to generate antimicrobial analogues of existing peptides

The performance of HydrAMP in analogue generation was compared to three alternative models: Basic, PepCVAE[29], and Joker[23]. HydrAMP was tested using three temperature parameter setups: a conservative $\tau = 1$, and more explorative $\tau = 2$ and $\tau = 5$ levels. Basic is a standard cVAE, but with the latent reconstruction regularization added, while PepCVAE is one of the state-of-the-art approaches to peptide

generation using the conditional variational autoencoder framework. PepCVAE lacked available code and was re-implemented by ourselves. Basic was adapted and implemented by us for AMP generation. In case of Basic, only reconstruction mode training was performed. For PepCVAE we used reconstruction and unconstrained modes. Both models lack analogue generation and Jacobian disentanglement regularization (see Methods for further details). Hence, the comparison of HydrAMP to these two models can be treated as an ablation study. Joker is a traditional model operating in the peptide sequence space, where a prototype peptide is given and Joker inserts the $\alpha$-helical patterns into peptide sequence, using a sliding window system. For running Joker, we used the code available at its GitHub repository.

In order to assess the analogue generation, we inspected the models' ability to obtain peptides with desired antimicrobial properties, as specified by setting the condition, in a number of different generation tasks. To this end, in each task, each generated peptide $p$ was assessed by the Classifier, and its probabilities of being AMP $\mathbb{P}_{M_{AMP}}(p)$ or being active $\mathbb{P}_{M_{MIC}}(p)$ were recorded.

First, we investigated the fraction of prototypes for which the model generated analogues that met the evaluation criteria. We defined two distinct evaluation criteria: *baseline discovery* and *improvement discovery*, with the former corresponding to the ability of simply generating analogues with good antimicrobial properties, and the latter corresponding to generating analogues with properties strictly better than the input prototype. Specifically, a peptide met the baseline discovery criterion if it had a probability of being AMP greater than 0.8 and probability of having low MIC greater than 0.5. A peptide met the improvement discovery criterion if its probability of being AMP and the probability of having low MIC were both greater than of the original peptide.

The models were asked to generate analogues of known, existing AMPs (referred to as *positives*; see Fig. 2a). 1319 AMP peptides from the test set (i.e. not used during training) were used as prototypes. In these experiments, each of the models: HydrAMP, Basic and PepCVAE was given a prototype sequence as input and the conditions were set to $c_{AMP} = 1$ (being antimicrobial) and $c_{MIC} = 1$ (being highly active against *E. coli*). Joker was run with the default parameters using the same set of peptides as input. In the case of baseline discovery task, we generated peptides for HydrAMP using Supplementary Algorithm 1 (Supplementary Methods 1.1), for PepCVAE, and Basic using Supplementary Algorithm 2 (Supplementary Methods 1.1), with the parameter `nb_of_tries`=64. In this task, novel analogues were accepted for at least 20% of input positive prototypes for all models. Depending on the temperature parameter, HydrAMP showed a 6, 31 and 45 percent point advantage in the fraction of accepted analogues compared to the next best model, PepCVAE. The acceptance rates for HydrAMP were 40% for $\tau = 1$; 65% for $\tau = 2$, and 79% for $\tau = 5$, respectively, and 34% for PepCVAE. Similarly, HydrAMP performed best in the improvement discovery task, regardless of the temperature value. In this task, HydrAMP was run using Supplementary Algorithm 3 (Supplementary Methods 1.2), while PepCVAE, and Basic were run using Supplementary Algorithm 4 (Supplementary Methods 1.2), with the parameter `nb_of_tries`=64. Here, based on the same set of prototype sequences, the HydrAMP model with $\tau = 5$ improved more than twice (1170) as many peptides as PepCVAE (452), almost four times more than the Basic model (301), and over four times more than Joker (280). These results show that HydrAMP model is the most creative of the tested models and has the capabilities of suggesting new analogues for original prototypes.

For further analysis of the analogue generation, we gave HydrAMP and the compared models a more challenging task of generating AMP analogues for 1253 known non-AMP peptides (negatives) from the test set (Fig. 2b). Unsurprisingly, meeting the baseline discovery criterion was much harder when the negatives were given as input compared to the positives. Still, HydrAMP managed to discover analogues for 85,

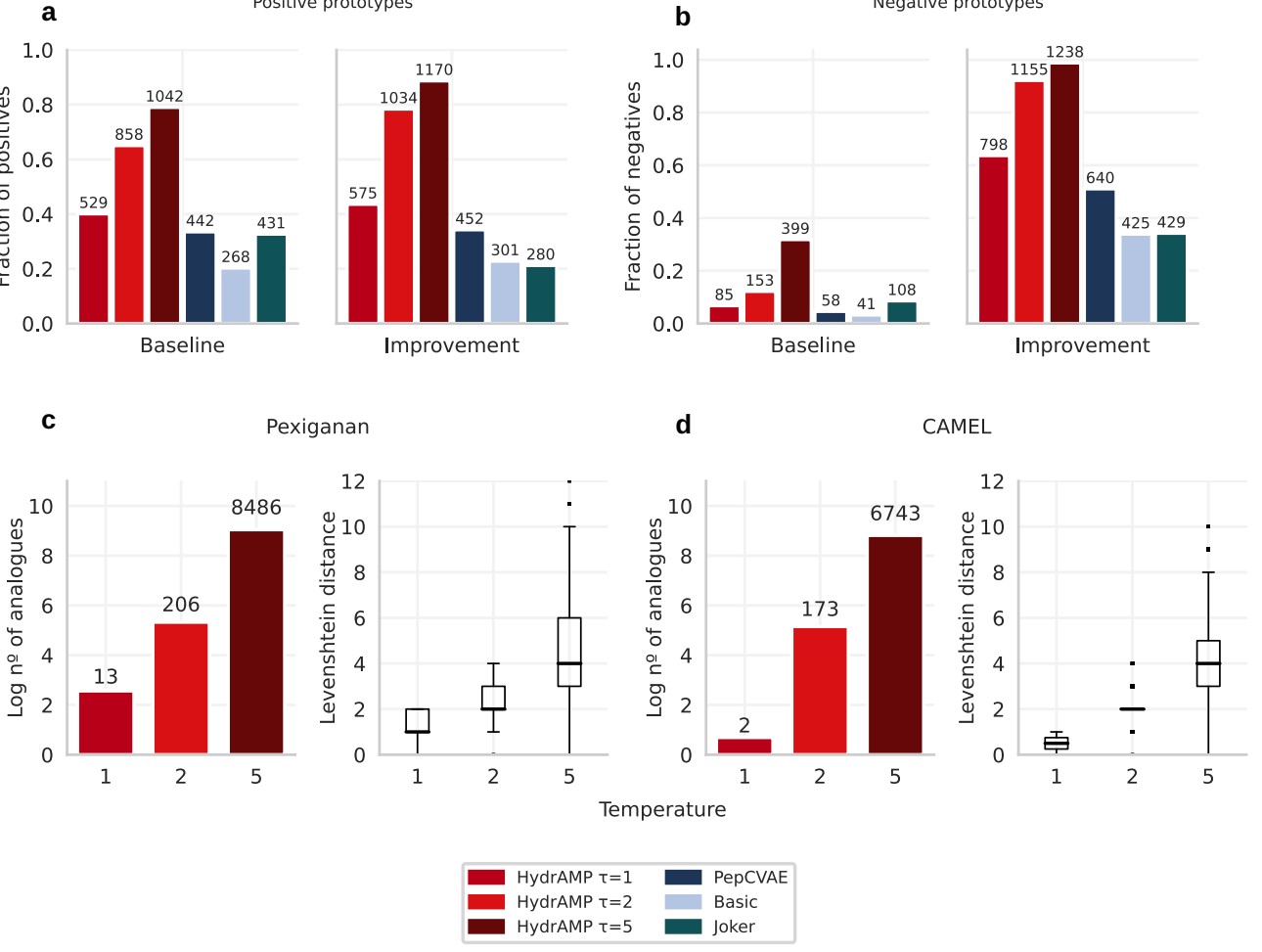

**Fig. 2 | Analogue generation performance in terms of number of generated analogues of HydrAMP (red), in comparison to PepCVAE (dark blue), Basic (light blue), and Joker (dark green). a** Fraction (y-axis) and number (over each bar) of 1319 positive (AMP and highly active) peptides from the test set, which produced analogues that met baseline discovery or improvement discovery criteria (x-axis) in the analogue generation. **b** As in **a**, but for 1253 negative peptides from the test set. **c, d** Left: The relation between the creativity parameter temperature ($\tau$; x-axis) and the log number of generated unique analogues that met the baseline discovery criteria, out of 10,000 total attempts (y-axis; the actual number of analogues shown above each bar). Right: the distribution of the Levenshtein distances between generated unique analogues out of 10,000 total attempts and the prototype sequence; for Pexiganan (**c**) and CAMEL (**d**). The borders of the box indicate the first quartile (bottom) and the third quartile (top) of the data. The line within the box indicates the median. The whiskers indicate the most extreme, non-outlier data points, whereas the dots behind the whiskers denote the outliers. The sample sizes for each box in (**c**) are $n = 13$ for $\tau = 1$, $n = 206$ for $\tau = 2$, and $n = 8486$ for $\tau = 5$. The sample sizes for each box in (**d**) are $n = 2$ for $\tau = 1$, $n = 173$ for $\tau = 2$, and $n = 6743$ for $\tau = 5$. Source data are provided as a Source Data file.

153, and 399 prototypes for three increasing temperature settings. Apart from the most conservative parameter $\tau = 1$, HydrAMP outperformed all competitor models in the baseline discovery task, providing over 1.4 (for $\tau = 2$), and 3.7 (for $\tau = 5$) times more discovered peptides than the top competitor Joker (108), and 3.7 (for $\tau = 2$) and 9.7 (for $\tau = 5$) times more than the worst competitor, Basic (41). For the most conservative $\tau = 1$ value, HydrAMP, with 85 discovered peptides, was outperformed only by Joker. The exceptional performance of HydrAMP for the less conservative, higher temperature values suggests that there are more changes needed to obtain a positive peptide starting from a negative one. Notably, HydrAMP was also able to improve almost 64%, 92%, or 99% of the test set negatives, depending on the temperature parameter. The best competitor, PepCVAE, improved only 51% (640) of the negative prototypes. For the least conservative temperature parameter $\tau = 5$, HydrAMP improved almost 3 times more peptides than the worst performing model, Joker.

In order to assess the creativity of HydrAMP model in the analogue generation, we evaluated the number of unique analogues created using two popular AMP peptides as prototypes: Pexiganan and CAMEL (Fig. 2c, d). The greater the temperature $\tau$, the more the generated analogues differ from the prototype. The ability to control the creativity of the model is important. The more similar a newly created analogue is to the prototype one, the more likely it is to preserve the physicochemical properties of the original peptide. On the other hand, the more alternations introduced, the larger the number of generated novel peptides.

In order to compare the antimicrobial properties of either positive or negative prototypes and their analogues that met either the baseline or improvement discovery criteria, we analyzed their probabilities of being AMP ($\mathbb{P}_{M_{AMP}}$) and having low MIC ($\mathbb{P}_{M_{MIC}}$; Fig. 3). We also compared these probability distributions against the distributions of all the peptides from the test set. We first investigated the negative prototypes and the analogues generated by HydrAMP, PepCVAE, Basic, and Joker that met the baseline discovery criterion. For $\tau = 5$, HydrAMP was able to produce analogues that met the baseline discovery criterion for prototypes that had lower a priori probability of being AMP (Fig. 3a) and lower probability of being highly active (Fig. 3b) than its competitors. In contrast, Basic and PepCVAE models produced analogues that met the baseline discovery criterion for peptides that were already initially likely to be antimicrobial and active. Joker's performance was

similar to HydrAMP's in terms of generating analogues for prototypes with very low $\mathbb{P}_{M_{MIC}}$ (Fig. 3b). However, HydrAMP with $\tau = 5$ was able to generate analogues for prototypes with much lower $\mathbb{P}_{M_{AMP}}$ (Fig. 3a), while the fraction of prototypes that met the evaluation criteria was much higher (recall Fig. 2b).

Next we investigated the $\mathbb{P}_{M_{AMP}}$ and $\mathbb{P}_{M_{MIC}}$ distributions for the negative prototypes and their analogues that met the improvement discovery criterion (Fig. 3c, d). For the negative prototypes this criterion puts less restrictive constraints on the resulting analogues compared to baseline discovery criterion. As negative prototypes have low $\mathbb{P}_{M_{AMP}}$ and $\mathbb{P}_{M_{MIC}}$ values, it will suffice that their analogues improve with respect to these low starting points without the need to reach $\mathbb{P}_{M_{AMP}} \geq 0.8$ and $\mathbb{P}_{M_{MIC}} \geq 0.5$ values. This investigation revealed that for all methods it is possible to improve $\mathbb{P}_{M_{AMP}}$ values of negative peptides to some extent (Fig. 3c). However, only HydrAMP for higher creativity values ($\tau \in \{2, 5\}$) achieved a strong mass shift of the $\mathbb{P}_{M_{AMP}}$ distribution towards high probability values for the obtained analogues. The $\mathbb{P}_{M_{MIC}}$ distribution also shifted towards higher values, but this effect was much less pronounced (Fig. 3d). Only HydrAMP for $\tau = 5$ and Joker managed to obtain a heavier tail of the distribution $\mathbb{P}_{M_{MIC}}$ around high values.

Furthermore, we evaluated $\mathbb{P}_{M_{AMP}}$ and $\mathbb{P}_{M_{MIC}}$ distributions for the positive prototypes and their analogues that met the baseline discovery criterion (Fig. 3e, f) and the improvement discovery criterion (Fig. 3g, h). For the baseline discovery criterion, we observed that while the positive prototypes already had high $\mathbb{P}_{M_{AMP}}$ and $\mathbb{P}_{M_{MIC}}$ values, for their analogues generated by all methods these values were even higher (Fig. 3e, f). Finally, for the improvement criterion, the $\mathbb{P}_{M_{AMP}}$ values for the prototypes were quite high (with median around 1), and all methods achieved a visible shift in their distribution towards even higher values (Fig. 3g). However, HydrAMP with high creativity parameter ($\tau = 5$) was the only method that managed to obtain $\mathbb{P}_{M_{AMP}}$ distribution for the generated analogues strongly concentrated around 1, resembling the one obtained in the baseline discovery. In contrast to the $\mathbb{P}_{M_{AMP}}$ values, the $\mathbb{P}_{M_{MIC}}$ values for the prototypes were quite low (with medians close to 0; Fig. 3h). Here, only HydrAMP with higher creativity parameters ($\tau \in \{2, 5\}$) and Joker were able to substantially shift the distribution of $\mathbb{P}_{M_{MIC}}$ towards high values (with median 1) for the analogues.

The exceptional performance of HydrAMP in generating highly antimicrobial and active novel peptides based on non-AMP prototypes shows its potential to provide truly novel antimicrobial peptides and increase the diversity of the pool of AMP sequences. Indeed, in contrast to analogues produced from positive prototypes, those peptides are expected to have sequences that largely differ from the sequences of known AMPs. In addition, the high performance of HydrAMP in improving positives promises the ability to generate novel AMPs with even higher activity than the currently available ones.

## HydrAMP outperforms other methods in unconstrained generation

We assessed the unconstrained generation abilities of HydrAMP in comparison to six other models: Basic, PepCVAE, as well as AMP-LM[34], Dean-VAE[27], Muller-LSTM[31], and AMP-GAN[28].

The models HydrAMP, Basic and PepCVAE were given the task of generating 50,000 peptides de novo in each of two modes: *positive* and *negative*. In the case of the *positive* mode we randomly sampled the latent representation vector and set the condition of the new peptide $p$ to be AMP ($c_{AMP} = 1$) and active ($c_{MIC} = 1$). In the negative mode, we set the peptide conditions to be non-AMP ($c_{AMP} = 0$) and not active ($c_{MIC} = 0$). In both of these tasks, HydrAMP was run using Supplementary Algorithm 5 (Supplementary Methods 2) with the parameter `nb_of_tries`=64, while PepCVAE, and Basic were run using Supplementary Algorithm 6 (Supplementary Methods 2).

The remaining four models (AMP-LM, Dean-VAE, Muller-LSTM, and AMP-GAN) do not have built-in functionality of setting the mode to generate positive or negative peptides. Their generated peptides are intended as positives. Hence, we evaluated these models solely in the positive mode. Specifically, AMP-LM is an LSTM-based language model generating sequences by predicting the next amino acid based on the previous amino acids in the sequence. For AMP-LM, we used peptides provided in Supporting Dataset 1 of[34] (*lstm.sample*) before any filtering was applied. Dean-VAE is a vanilla VAE with both encoder and decoder based on LSTMs. For Dean-VAE, we used available generated peptides with sampling step 'active' or '.99' from Table S2 of[27]. Muller-LSTM is another LSTM-based model for AMP design. For Muller-LSTM, we collected peptides from Supporting Information S3 of[31]. From all the downloaded datasets we removed sequences exceeding the 25 amino acids in length, sequences containing non-standard aminoacids, as well as dropped empty entries, leaving a total of 24,588, 2973, and 976 sequences for AMP-LM, Dean-VAE and Muller-LSTM, respectively. AMP-GAN is a conditional GAN with a latent consisting of binary codes of peptide structure and peptide properties. We used the trained model from the GitHub repository and generated 100,000 sequences. We removed the sequences longer than 25 amino acids and randomly truncated the datasets to 50,000 sequences needed for fair comparison with HydrAMP. In this evaluation we did not compare to Joker[23] as it is not a suitable method for unconstrained generation.

Peptides generated by HydrAMP were more likely to follow the desired conditions compared to the the competing models both for AMP (Fig. 4a, b) and MIC (Fig. 4c, d) conditions. The advantage of HydrAMP was the most noticeable in the case of the probability $\mathbb{P}_{M_{MIC}}$ of having low MIC: the median probability of having low MIC for peptides generated by HydrAMP was around 0.75, while for the next best model, AMP-LM, it was around 0.1 (Fig. 4c).

In order to confirm that HydrAMP is able to suggest new, active AMP peptides when using the unconstrained generation we tested to what degree the generated peptides match appropriate criteria (Fig. 4e). For that, we generated 50,000 candidate peptides and run three experiments confirming their properties. First, we computed the fraction of such peptides $p$ which had high probability of being AMP ($\mathbb{P}_{M_{AMP}} > 0.8$). Second, we computed the fraction of peptides with high probability of being active ($\mathbb{P}_{M_{MIC}}(p) > 0.5$). Finally, we computed the fraction of peptides which have both high probability of being AMP and high probability of being active. We compared HydrAMP's performance with AMP-GAN, as this was the only method that provided enough candidate peptides for the subsequent filtering steps.

HydrAMP suggested the largest number of peptides that had high probability of being AMP, high probability of having low MIC against *E. coli*, and that met both those criteria, i.e. were confirmed as positives. Eventually, HydrAMP created almost 44% more peptide candidates that matched all three conditions than its competitor AMP-GAN.

## HydrAMP generates peptides with desired physicochemical properties

Next, we evaluated HydrAMP by inspecting the physicochemical properties of peptides that it generates in analogue generation, in comparison to the properties of known peptides, as well as peptides generated by PepCVAE, Basic, and Joker[23] (Fig. 5). Physicochemical properties of antimicrobial (positive) peptides differ from peptides that were experimentally verified not to be antimicrobial (negative peptides). Indeed, a comparison of the distributions indicated that isoelectric point (Fig. 5a), charge (Fig. 5d), hydrophobic ratio (Fig. 5g), and aromaticity (Fig. 5j) are significantly larger (one-sided Mann-Whitney test $p$ value < 0.05) for known positives than for negatives.

We first inspected the performance of HydrAMP and other approaches in the task of finding positive peptides based on negative prototypes in the analogue generation. Given such significant

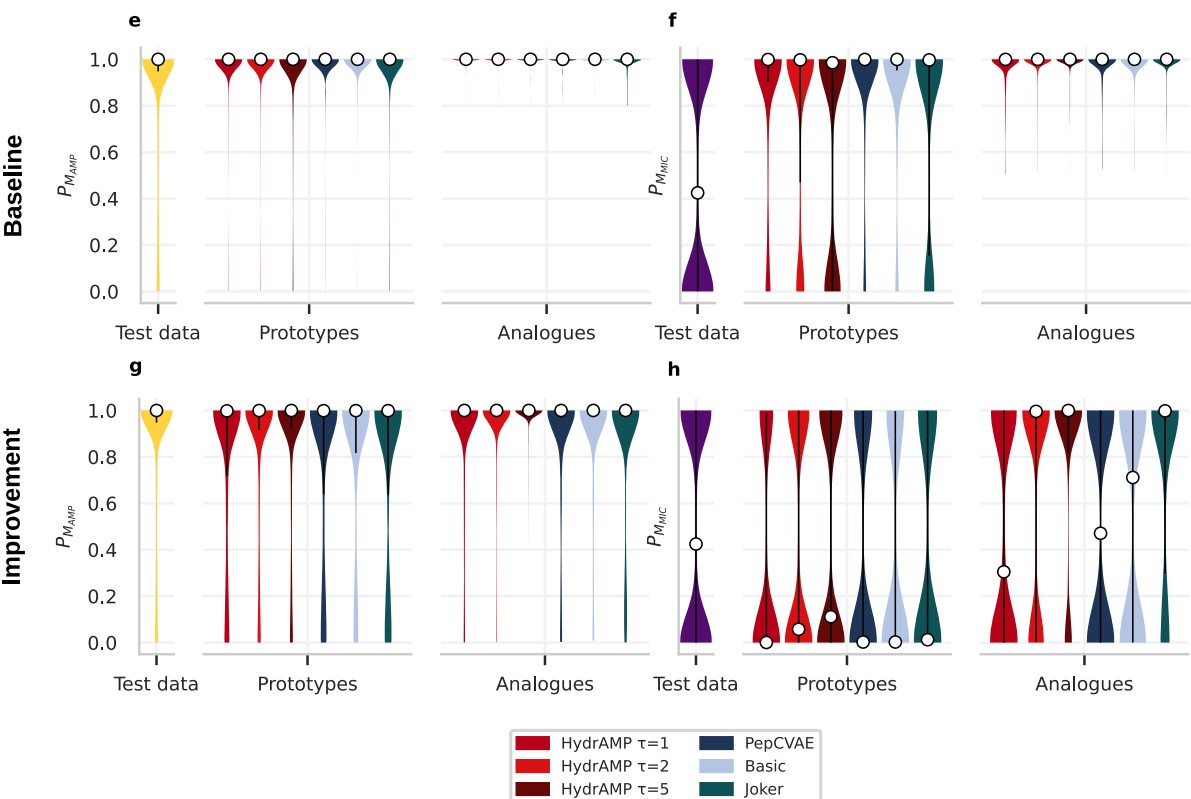

Fig. 3 | Analogue generation performance in terms of $\mathbb{P}_{M_{AMP}}$ and $\mathbb{P}_{M_{MIC}}$ for HydrAMP (red), in comparison to PepCVAE (dark blue), Basic (light blue), and Joker (dark green) compared to the distribution of test data (yellow for $\mathbb{P}_{M_{AMP}}$ and violet for $\mathbb{P}_{M_{MIC}}$). Test data contains all the prototypes and serves as a reference. **a, b, c, d** The probability distributions of $\mathbb{P}_{M_{AMP}}$ (**a, c**) and $\mathbb{P}_{M_{MIC}}$ (**b, d**) for $n = 1253$ negative prototypes and the generated analogues that met the baseline (**a, b**) or improvement (**c, d**) discovery criteria. **e, f, g, h** The probability distributions of $\mathbb{P}_{M_{AMP}}$ (**e, g**) and $\mathbb{P}_{M_{MIC}}$ (**f, h**) for $n = 1319$ positive prototypes and the generated analogues that met the baseline (**e, f**) or improvement (**g, h**) discovery criteria. The white dots mark the median of each distribution. The black vertical lines denote the interquartile range of each distribution. Source data are provided as a Source Data file.

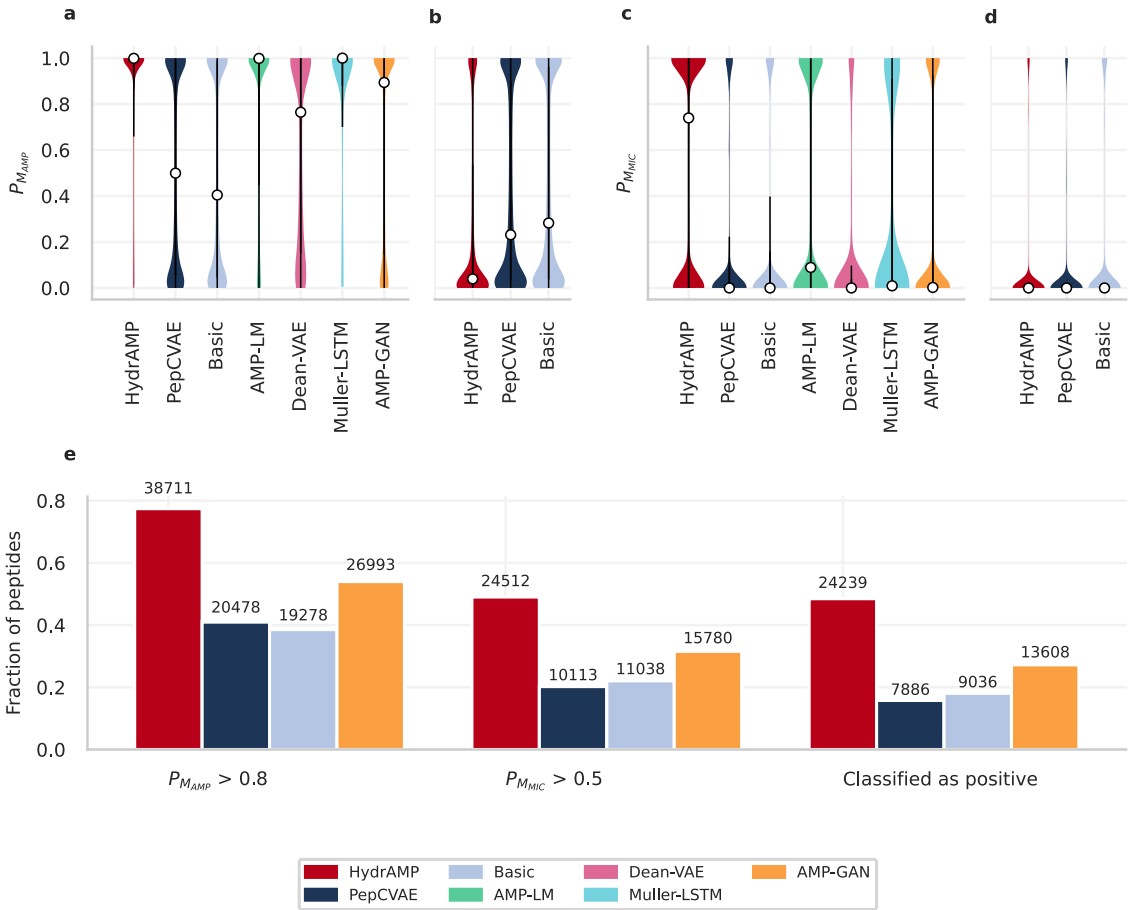

**Fig. 4 | Unconstrained generative performance of HydrAMP (red), in comparison to PepCVAE (dark blue), Basic (light blue), AMP-LM (green), Dean-VAE (pink), Muller-LSTM (aquamarine), and AMP-GAN (orange) (methods indicated on the x-axis). a, b** The distribution of probabilities of being antimicrobial ($\mathbb{P}_{M_{AMP}}$), for the task of generating positive peptides (**a**) and for the task of generating negative peptides (**b**). **c, d** The distribution of probabilities of being highly active (having low MIC, $\mathbb{P}_{M_{MIC}}$) for the task of generating positive peptides (**c**) and for the task of generating negative peptides (**d**). In panels **a, b, c, d**, the white dots mark the median of each distribution. The black vertical lines denote the interquartile range of each distribution. Sample sizes in panels **a**, **c**: HydrAMP $n = 50,000$, PepCVAE $n = 50,000$, Basic $n = 50,000$, AMP-LM $n = 24,588$, Dean-VAE $n = 2973$, Muller-LSTM $n = 976$, AMP-GAN $n = 50,000$. Sample sizes in panels **b**, **d**: HydrAMP, PepCVAE and Basic $n = 50,000$. **e** For the task of generating positive peptides, fraction of generated peptides with $\mathbb{P}_{M_{AMP}} > 0.8$ (first bar plot), fraction of peptides with $\mathbb{P}_{M_{MIC}} > 0.5$ (second bar plot), fraction of peptides that satisfy both previous criteria i.e. classified as positive (third bar plot). The number over each bar: the actual number of peptides that met the condition. Source data are provided as a Source Data file.

differences between negatives and positives, this requires the introduction of a shift in the physicochemical properties from the non-AMP prototypes to the newly generated peptides. Here, HydrAMP was once again tested with three different temperature setups: $\tau = \{1, 2, 5\}$. For $\tau = 2$ and $\tau = 5$, HydrAMP showed a capacity to generate analogues with the desired significant increase (one-sided Mann-Whitney test $p$-value $< 0.05$) of all investigated properties: isoelectric point (Fig. 5b), charge (Fig. 5e), hydrophobic ratio (Fig. 5h), and aromaticity (Fig. 5k). For these higher temperature levels, the shift in all the physicochemical properties was much stronger than for Basic and PepCVAE models. This implies that we can control the balance between the shift in physicochemical properties and the degree of changes between new analogues and their prototypes. In contrast to HydrAMP, all four physicochemical properties of peptides generated by PepCVAE or Basic showed no significant difference to the properties of known negatives. We confirmed that these results could not be obtained by chance by computing physicochemical properties of a randomly sampled subset of the UniProt dataset and peptides for which we randomly sampled a sequence of amino acids. In both of these cases, HydrAMP in explorative temperature setups produce peptides with better qualities. While both HydrAMP and Joker can produce significant shift in physicochemical properties between the negative

prototypes and the generated analogues, HydrAMP is able to suggest analogues for all the prototypes (1253), as opposed to Joker (556), regardless of temperature. The performance of HydrAMP in the conservative temperature setting $\tau = 1$ confirms that generating a positive analogue based on negative prototype requires a larger degree of changes. However, even for such conservative temperature HydrAMP is able to generate peptides with a significant shift of isoelectric point (Fig. 5b), charge (Fig. 5e), hydrophobic ratio (Fig. 5h) as compared to those in the negative set.

Second, we evaluated HydrAMP and other approaches in a task of improving the positives. Here the challenge is different than in the previous task, as it is hard to improve peptides that are already "good" (are already AMP and active). Here, for all analyzed physicochemical properties, HydrAMP in explorative temperature setups $\tau = \{2, 5\}$ generated peptides with values of physicochemical descriptors significantly higher values (one-sided Mann-Whitney test $p$ value $< 0.05$) than those expected for active peptides (Fig. 5c, f, I, l). The benefit of the model's creativity using a larger temperature parameter is most visible for aromaticity (Fig. 5l). In this task, Joker's performance is insufficient to improve positives in terms of isoelectric point (Fig. 5c) or aromaticity (Fig. 5l). Additionally, HydrAMP produces analogues for all of the positive prototypes (1319), which is not the case with Joker

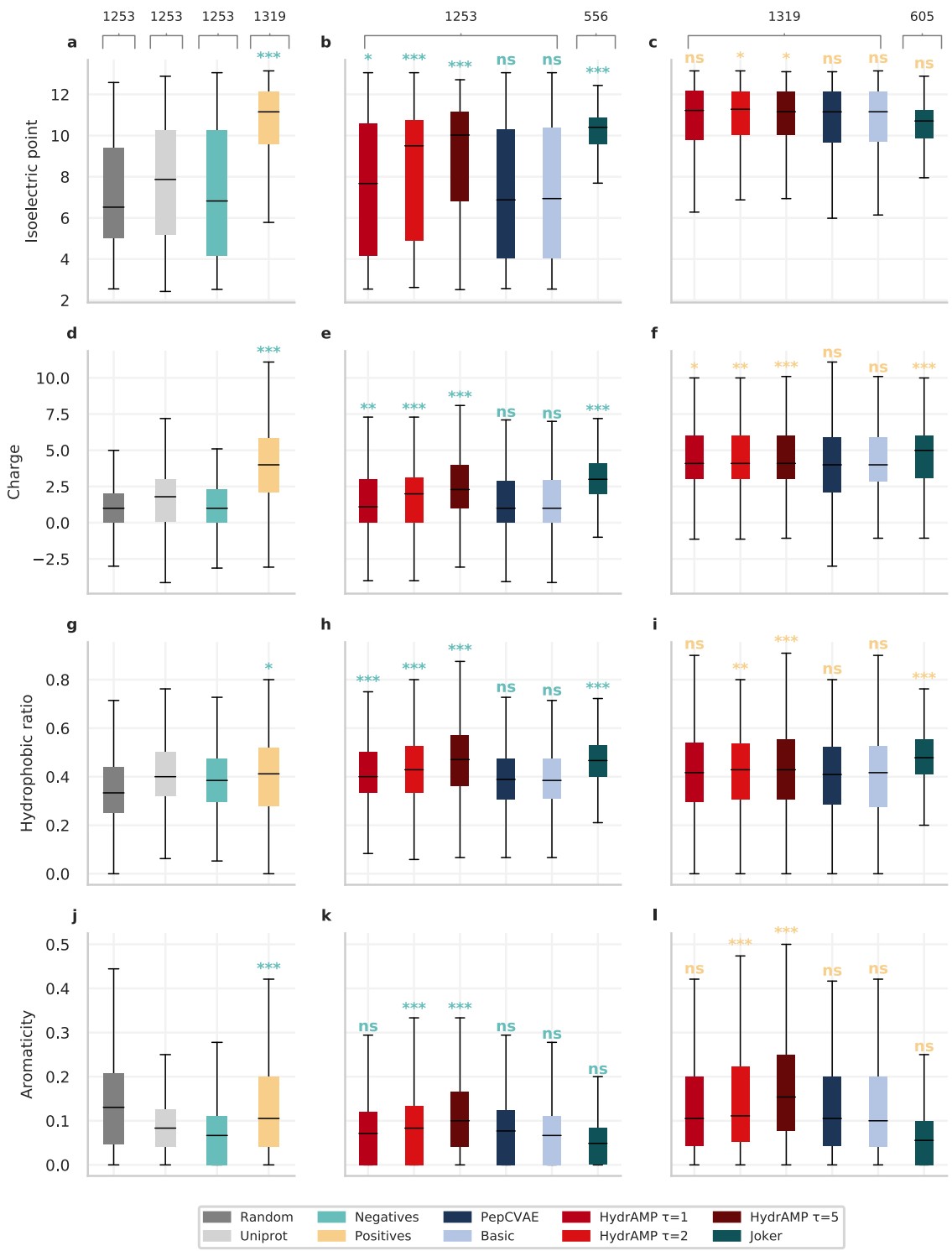

**Fig. 5 | Physicochemical properties of analogues generated by HydrAMP and compared methods in analogue generation for non-AMP or AMP prototypes in comparison with real and random data.** Distributions of properties (**a, b, c** Isoelectric point, **d, e, f** Charge, **g, h, i** Hydrophobic ratio, **j, k, l** Aromaticity) of randomly generated peptides (dark gray; sample size $n = 1253$;), peptides sampled from UniProt (light gray; $n = 1253$), true negatives (green; $n = 1253$), and true positives (yellow; $n = 1319$), in comparison with AMP analogues generated from negatives (**b, e, h, k**), and positives (**c, f, i, l**) by different models: HydrAMP with various creativity parameter *temperature* values: $\tau = 1$ (light red), $\tau = 2$ (red), $\tau = 5$ (dark red), PepCVAE (dark blue), Basic (light blue), and Joker (dark green). HydrAMP generated

analogues for $n = 1253$ negatives (for all temperature values), Joker generated analogues for $n = 556$ of them. For the positive test set, HydrAMP generated analogues for all $n = 1319$ peptides (for all temperature values), while Joker generated analogues for $n = 605$ of them. The significance levels of one-sided Mann-Whitney test are denoted above the boxes as: ns - $P \geq 0.05$; * - $P \leq 0.05$; ** - $P \leq 0.01$; *** - $P \leq 0.001$. The borders of the boxes indicate first quartile (bottom) and the third quartile (top) of the data. The line within the box indicates the median. The whiskers indicate the the most extreme, nonoutlier data points. Source data are provided as a Source Data file.

(605). PepCVAE and Basic fail to produce peptides with significantly higher values than those expected for active peptides in all four of the considered physicochemical properties.

Finally, we evaluated the physicochemical properties of positive and negative peptides generated by HydrAMP in the unconstrained mode, in comparison to peptides generated by other methods (Supplementary Fig. 1). For all four analyzed physicochemical properties, there was large and significant difference between their distributions for generated negatives and positives generated by HydrAMP (one-sided Wilcoxon test $p$ value < 0.05). For PepCVAE and Basic, the differences between negatives and positives were also significant, but the medians of the distributions did not differ by as large amount as for HydrAMP. We could not include AMP-GAN[28] in the comparison between generated positives and negatives as it lacks the functionality of generating negative peptides. Other methods were not applicable because they provided too few generated peptides.

We confirmed that the amino acid distribution of the peptides obtained by HydrAMP in the unconstrained generation is in agreement with the true amino acid frequencies (Supplementary Fig. 2). High content of lysine (K) and arginine (R), as well as low content of negatively charged glutamic acid (E) and aspartic acid (D) contribute to the positive net charge.

Overall, these results illustrate the superior performance of HydrAMP in generating peptides with desired physicochemical properties, reflecting the properties observed for real peptides.

## HydrAMP suggested valid novel peptides without targeted preselection of candidates based on auxiliary classifiers or MD simulations

Next, we inspected the ability of HydrAMP to generate novel antimicrobial peptides using experimental wet lab validation. In this experimental round, we intended to test peptides without any additional, classifier- and simulation-based preselection. To this end, we applied HydrAMP in the analogue mode, with temperature (creativity) parameter $\tau = 1$, treating Pexiganan and Temporin-A as prototypes (see Supplementary Data 2 for the experimental characteristics of the prototypes). The low creativity parameter choice was justified by the fact that we intended to obtain novel peptides that were similar to the prototypes. For each prototype, we first generated a set of 900 positive analogues. Next, we applied filtering criteria to both generated sets of peptides (see Biological filtering criteria) to increase the chance of peptides being synthesizable and were left with 92 sequences for Pexiganan and 84 for Temporin-A. Out of these candidates, four positive analogues of Pexiganan and four of Temporin-A were handpicked by experts for experimental validation. One of the selected Pexiganan analogues did not synthesize. The remaining were investigated experimentally and their antimicrobial activity was tested against two *E. coli* strains (Table 1; Methods). Laboratory experiments validated the known and predicted by the Classifier very high antimicrobial activity of Pexiganan ($\mathbb{P}_{M_{AMP}} = 0.99, \mathbb{P}_{M_{MIC}} = 0.99$). Out of three synthesizable analogues of Pexiganan, which were generated as positive (with conditions fixed to ($c_{AMP} = 1, c_{MIC} = 1$)), one showed even higher activity than that of Pexiganan (MIC = 2 μg/mL). This novel validated antimicrobial peptide was called Hydraganan-1. Another predicted analogue was confirmed as AMP and of high activity (MIC = 16 μg/mL) and was named Hydraganan-2.

The Classifier model predicted high probability of being antimicrobial, and a very low probability of being highly active for Temporin-A ($\mathbb{P}_{M_{AMP}} = 0.99, \mathbb{P}_{M_{MIC}} < 0.01$). The low activity was confirmed in the lab (MIC = 256 μg/mL). All four generated positive analogues of Temporin-A also showed low antimicrobial activity (MIC ≥ 512 μg/mL). As controls, we randomly selected and validated two peptides generated as negative analogues by HydrAMP, one for Pexiganan, and one for Temporin-A. Both negative analogues were validated as inactive (MIC > 512 μg/mL).

Taken together, from the total seven peptides that were generated by HydrAMP as positives for the two prototypes and tested, two were validated as positives, i.e. active against *E. coli*. This result corresponds to a positive validation rate of 29%

## Simulations confirmed that Hydraganan penetrates the cell membrane

To better understand the functional properties of Hydraganan and the experimentally validated negative analogue of Pexiganan, Pex-P1-4, we performed extensive, fully atomistic simulations of their behavior in the presence of a lipid membrane (Fig. 6a, b). Each peptide was constructed as a regular α-helical structure and placed in aqueous compartment of the simulation box with its helical axis parallel to the membrane plane, at three different orientations. During the simulations we monitored the stability of peptides α-helical structure as well as the degree of their association with the membrane.

Notably, the active Hydraganan-1 consistently maintained ~ 0.75 fraction of helical geometry, that is considerably more than in the case of the inactive analogue, whose helical structure content dropped well below 0.5 already within the first 200 ns (Fig. 6e, upper plot). It has been reported that many AMPs with stable α-helical scaffold show properly aligned hydrophobic surface patch on one peptide side and oppositely facing cluster of hydrophilic or charged side chains[37]. While the former provides a driving force for peptide insertion into the nonpolar membrane core, the latter apparently prevents membrane defect from closing. Notably, these features rely not merely on individual amino acids but rather on their appropriate distribution within the sequence that warrants correct placement upon folding as well as helical propensity of the resulting structure.

The degree of association of a peptide with the membrane was assessed by introducing a parameter $S \in [0, 1]$ that reflects the fraction of peptide heavy atoms buried below the membrane surface (Fig. 6d), with $S = 0$ indicating no association, while $S = 1$ representing full peptide burial (see Computer simulations of peptide-membrane systems). The evolution of $S$ during simulation time was monitored by calculating it over consecutive 25 ns intervals. We emphasise that the $S$ parameter is indicative of the peptides' ability to bind and penetrate the membrane core, which is a necessary but not sufficient condition to warrant antimicrobial activity. Moreover, the binding itself and its further biological effect, can be modulated by a number of factors such as pH, salt nature and concentration, additional components of bacterial cell envelope, or cooperation between multiple AMP units, which were not included in the simulation model. In the case of Hydraganan-1, irrespective of initial orientation, we observed peptide association with the membrane within the first 250 ns of the simulation time, as indicated by $S$ values increasing above 0.5 (Fig. 6e, lower plot). In all cases the association was followed by complete peptide burial ($S \to 1$) within ~ 400 ns. In contrast, the negative Pexiganan analogue, Pex-P1-4, only loosely adhered to lipid bilayer surface and revealed no tendency to penetrate into its core, with $S$ values remaining around 0.25 till the end of simulations. Despite its limitations, the favourable agreement of $S$ with membrane activities expected for Hydraganan-1 and Pex-P1-4 suggests that it may have a discriminative power in distinguishing between positive and negative peptides.

## Auxiliary classifier-based and simulation-based evaluation as efficient preselection criteria for potent AMP candidates

While HydrAMP was able to propose active peptides without auxiliary, post hoc screening of candidates (Table 1), we introduced an additional preselection procedure aiming at selecting the most promising candidates for experimental validation, thereby limiting its cost. The preselection procedure, apart from the previously applied criteria (Biological filtering criteria), employed consensus filtering and

**Table 1 | Pexiganan and Temporin-A analogues obtained in the analogue generation process without additional preselection**

| ID | Name | Sequence | Condition (c_AMP, c_MIC) | MIC *E. coli* ATCC 43927 µg/mL | MIC *E. coli* ATCC 25922 µg/mL |
|---|---|---|---|---|---|
| 0 | ***Pexiganan*** | GIGKFLKKAKKFGKAFVKILKK | | **4** | **4** |
| 1 | Hydraganan-1 | GVGKKLWFALKKPLGLVKFFKLL | (1,1) | **2** | **2** |
| 2 | Hydraganan-2 | GIGKFLKFALKKGLGLVLKFKL | (1,1) | **16** | **16** |
| 3 | Pex-P1-3 | GVAKKLWIAAKKPAGAGSKFKLL | (1,1) | 512 | >512 |
| 4 | Pex-P1-4 | GELKKLWQAGKLSEEDGGAFKAG | (0, 0) | >512 | >512 |
| 0 | ***Temporin-A*** | FLPLIGRVLSGIL | | 256 | 256 |
| 1 | Temp-P1-1 | FLPLIGRVFSGIL | (1,1) | 512 | 512 |
| 2 | Temp-P1-2 | FLPLIGRVFSGIK | (1,1) | >512 | >512 |
| 3 | Temp-P1-3 | FLPLIGRVLSGIA | (1,1) | 512 | 512 |
| 4 | Temp-P1-4 | FLPLIGRVKSGIK | (1,1) | >512 | >512 |
| 5 | Temp-P1-5 | FLPIKNRYASAAE | (0, 0) | >512 | >512 |

Each row corresponds to a single peptide. Minimal Inhibitory Concentration (MIC) (µg/mL) was measured for each peptide against reference strains of microorganisms (*E. coli* ATCC 43927, *E. coli* ATCC 25922). Bold was applied to the name of the prototypes (Pexiganan, Temporin-A). The analogues with underlined names demonstrate activity in agreement with the condition (c_AMP, c_MIC) set during generation. Values of MIC measurements with activity MIC ≤ 32 µg/mL indicating a high antimicrobial activity are written in **underlined bold**.

preranking based on combined predictions of auxiliary classifiers, followed by fully atomistic molecular simulations (Supplementary Fig. 3).

The design of the preselection procedure was preceded by a comprehensive evaluation of 10 published AMP classifiers[6,8,38–42] with respect to their ability to correctly indicate peptides showing high activity against *E. coli*. The assessment was performed on a dedicated dataset that included as positives such peptides that were experimentally verified as active against *E. coli* (Supplementary Table 2). As no single classifier outcompeted others in this task (Supplementary Methods 3.1, Supplementary Table 3), we conservatively opted for using classifier consensus as a step of the preselection procedure to maximize the probability of filtering out false positive candidates. Next, we compared the classifiers by their ability to result in reliable ranking of top AMP candidates (Supplementary Methods 3.2; Supplementary Fig. 4). Based on this analysis, three best-performing models were selected (AMPlify[6], CAMPR3-rf[40] and STM[39]) to inform the ranking step of the preselection procedure. To optimize the ranking based on the predictions of the three selected classifiers, we next determined a set of weights with which the predictions of AMPlify, CAMPR3-rf and STM were combined for the ranking score (Supplementary Methods 3.3). Since toxicity classifiers showed relatively weaker performance (Supplementary Methods 3.4), we did not include those in the preselection procedure.

Finally, we devised a fully atomistic simulation protocol that would aid the preselection procedure in the assessment of peptides' membrane binding ability and potential activity. Compared to the simulations used to analyze Hydraganan-1 and the negative analogue Pex-P1-4 discussed above (Fig. 6a, b) this protocol was optimized in terms of efficiency and reliability, and consisted of two rounds (Fig. 6c, f). In the first round, a given peptide, with a starting geometry of regular α-helix, was placed in aqueous compartment, 3 nm from the membrane surface (for details see Methods), and was allowed to diffuse freely for 1 µs. If it achieved an average value of $S > 0.5$ within the last 100 ns of the MD run, it was considered capable of membrane binding and presumably active. Otherwise, it was subjected to the second round, in which the same starting structure was placed and equilibrated 0.3 nm from the membrane surface, and was left for 500 ns of unconstrained simulation. Here, the peptide was presumably active, if an average $S > 0.5$ was maintained within the last 100 ns. Lastly, the greater of the two obtained average $S$ values was assumed to represent peptide's activity for ranking purposes.

We validated the discriminative power of the protocol by running it on a set of peptides with experimentally verified activity (4 active, 1 moderately active, and 4 inactive). The MD simulations showed agreement with this classification in 8 out of 9 cases (Supplementary Fig. 5, Supplementary Data 3). According to our observations, the average content of α-helical secondary structure turned out to be a less reliable indicator of peptide activity. At least in part, it may result from the fact that similar degree of partial unfolding has different impact on membrane binding for shorter and longer amino acid chains.

## HydrAMP with additional targeted preselection of potent candidates suggested a large number of valid novel peptides with high antimicrobial activity

The framework combining HydrAMP with the additional preselection procedure was next used to indicate the most promising AMP candidates. This time we aimed at identifying analogues of known AMPs that were relevant in the context of clinical trials: OP-145[43], Omiganan[44], and Syphaxin[45] (Supplementary Data 2). As an additional challenge, we selected GQ20[46] as an experimentally validated, inactive peptide, with the intention to generate its improved, highly active analogues. Since HydrAMP is the only existing generative model of peptides that is trained for the improvement of negatives, this validation was intended as a proof of concept demonstrating the power of our approach.

Specifically, we proceeded according to the following workflow (Supplementary Fig. 3): we first applied HydrAMP to generate a number of candidates per each of the temperature $\tau \in \{1, 2, 5\}$ values, and per each of the four prototypes (Supplementary Table 4). We next removed duplicated sequences that occurred for different temperatures, and those that did not meet basic biological synthetizability criteria (see Biological filtering criteria). Next, we applied the consensus filtering step, retaining only such candidates for which all 10 auxiliary classifiers agreed that the peptide was AMP with high probability, and we ranked them according to the optimized weighted score (see Supplementary Methods 3.3). The selected top-generated peptides for each prototype (180 peptides in total) were then subjected to the optimized MD simulation protocol in order to evaluate their $S$ values as indicators of antimicrobial activity (Supplementary Table 4, Supplementary Data 4). Finally, we re-ranked the peptides by the $S$ values and submitted six top peptides for each prototype for experimental validation.

All of the submitted peptides were synthesized and their MIC and MBC were determined against five microbial strains: *Staphylococcus aureus* (*S. aureus*) ATCC 33591, *E. coli* ATCC 25922, *E. coli* ATCC 43827, *Acinetobacter baumannii* (*A. baumannii*) ATCC BAA 1605, *Pseudomonas aeruginosa* (*P. aeruginosa*) ATCC 9027. This set of strains was extended compared to the first experimental round. It contains two additional gram-negative species, as well as a gram-positive bacteria. Such a selection allowed us to compare activity of tested peptides against *E. coli* to other gram-negative bacteria, as well as activity

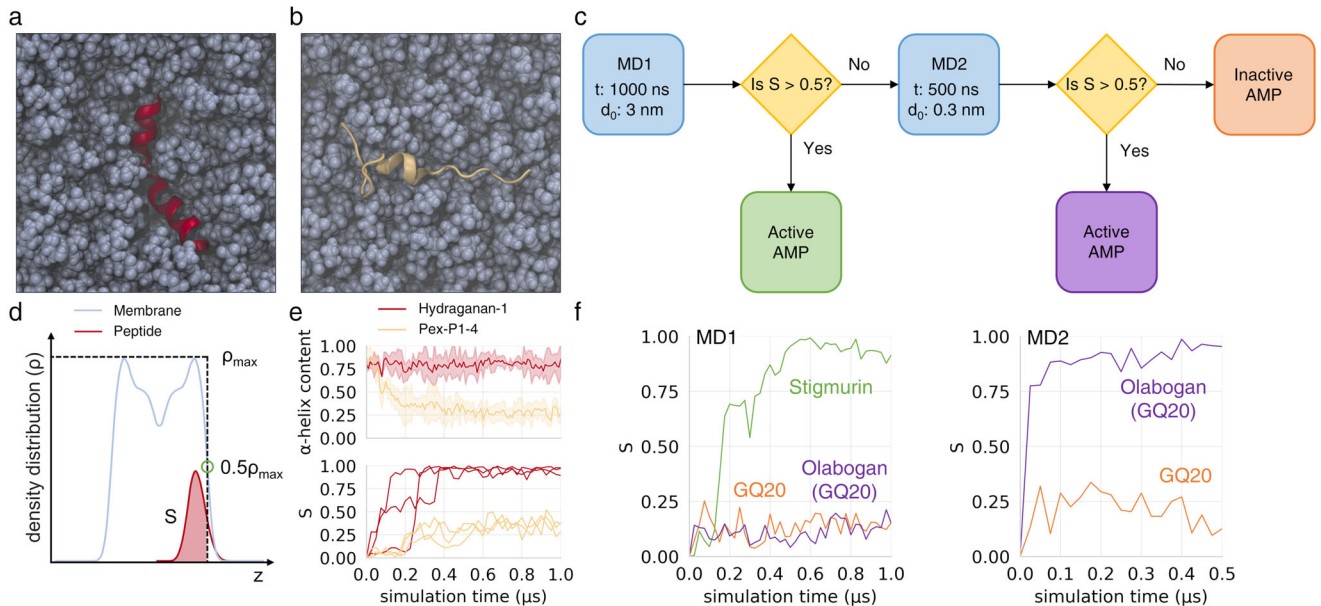

**Fig. 6 | Summary of atomistic molecular dynamics simulations of peptide-membrane systems. a, b** Late simulation snapshots of experimentally verified AMP Hydraganan-1 (red) and non-AMP peptide Pex-P1-4 (beige), respectively, in the membrane (blue); top view on membrane surface; water molecules not depicted for clarity; **c** scheme of optimized simulation protocol used for AMP preselection; $t$ -- simulation time; $d_0$ initial peptide-membrane separation; **d** scheme illustrating the evaluation of the $S$ parameter that describes the level of peptide burial within the membrane; $z$ -- membrane normal axis; **e** system descriptors in Hydraganan-1 and non-AMP Pex-P1-4 simulations; upper plot: an average (lines) and standard deviation (shaded areas) of $\alpha$-helix fraction within peptide residues; lower plot: the evolution of $S$ parameter for each of three initial peptide placements; **f** time evolution of $S$ parameter for three sample peptides illustrating possible routes in the preselection algorithm. The time evolution suggests that Stigmurin and Olabogan are active AMPs, while GQ20 is inactive. Source data are provided as a Source Data file.

between gram-negative and gram-positive species. The selected bacteria are highly clinically relevant. In particular, *S. aureus* ATCC 33591 is methicillin-resistant[47], *A. baumannii* ATCC BAA 1605 is resistant against Gentamicin, Ciprofloxacin, and Imipenem, among many other antibiotics[48].

The obtained MIC values indicated overall high activity of the generated peptides. The MBC measurements generally agreed with and were only rarely larger than the obtained MIC values, confirming the high activity (Table 2, see Supplementary Data 5 for additional information, such as predictions of auxiliary classifier, $S$ score, and physicochemical properties of these peptides). A tested peptide was considered an AMP when it was active against at least one of the tested strains. For a given activity threshold, we computed the *AMP success rate* as the fraction of peptides considered AMP according to this threshold out of all tested candidates. We compared this AMP success rate to success rates computed based on MIC measurements reported for the Joker model by Porto et al.[23] and for the CLaSS model by Das et al.[30] (see Supplementary Data 6 for computations). The CLaSS model is based on the Wasserstein Autoencoder and is equipped with additional classifiers assessing typical AMP properties such as activity and toxicity that guide rejection sampling of the generated peptides. Additionally, it filters the generated peptides using molecular dynamics-based descriptors. The Joker model filtered the generated peptides using biological pre-screening using the SPOT spot assay.

Previous studies located the activity threshold at around 128 $\mu$g/mL[23,30]. With this permissive activity threshold, as many as 23 out of 24 peptides generated by HydrAMP were confirmed as AMP. This resulted in AMP success rate of 96%, a bit higher than 11 out of 12 (92%) reported for the Joker model by Porto et al.[23]. In comparison, for this threshold, the CLaSS model by Das et al.[30] reported an AMP success rate of only 14%, obtaining only 3 active peptides out of 21 candidates that were tested.

In fact, more conservative activity thresholds should be considered more clinically relevant, as lower inhibitory concentrations should be easier to administer and be more cost-efficient. For a range of lower MIC thresholds of activity (8, 16, 32, and 64 $\mu$g/mL; Fig. 7), HydrAMP obtained higher AMP success rates than the compared methods, with the only exception of the threshold 32 $\mu$g/mL, where it was slightly exceeded by Joker. In particular, the AMP success rate of HydrAMP for the very high activity (8 $\mu$g/mL) was around 4.5 times higher than the success rate of Joker and almost 8 times higher than the success rate of CLaSS. The MIC thresholds 2 and 4 $\mu$g/mL were not considered in this analysis as such low values were not reported for neither by Joker nor CLaSS models. This analysis demonstrated exceptional generative performance of HydrAMP, especially its ability to propose highly active peptides.

In addition, we compared the AMP success rates of HydrAMP and the model of Ma et al.[16]. This model can be considered as a representative of the QSAR methods, and reported a success rate of 84%. This result was obtained by testing 216 peptides, 181 of which obtained MIC ≤ 60 $\mu$M. Note that this study used different MIC units and tested only a single inhibitory concentration. Hence it could not be included in the comparative analysis for different MIC thresholds above. After conversion of our MIC measurements to these units (Supplementary Data 7) we observed that the peptides showing MIC of 128 $\mu$g/mL had corresponding recalculated values of around 60 $\mu$M, reconfirming that the commonly used activity threshold is permissive. In total, 23 out of 24 peptides generated by HydrAMP had the recalculated MIC ≤ 60 $\mu$M, corresponding to a higher AMP success rate (96%) than the 84% reported by Ma et al.[16].

Additional tests of the previously found Pexiganan analogues, Hydraganan-1 and Hydraganan-2, showed their activity against more bacteria than only *E. coli*. In particular, Hydraganan-1 proved highly active (MIC ≤ 32 $\mu$g/mL) against all five tested strains of bacteria, while

**Table 2 | Experimentally validated analogues of GQ20, Syphaxin, OP-145, and Omiganan obtained through analogue generation with additional targeted preselection**

| ID | Name | Sequence | HC50 | S. aureus ATCC 33591 | | E. coli ATCC 25922 | | E. coli ATCC 43827 | | A. baumannii ATCC BAA 1605 | | P. aeruginosa ATCC 9027 | |
|---|---|---|---|---|---|---|---|---|---|---|---|---|---|
| | | | | MIC | MBC | MIC | MBC | MIC | MBC | MIC | MBC | MIC | MBC |
| 0 | *GQ20* | GQLNKFIKKAQRKFHEKFAK | >512 | >512 | >512 | >512 | >512 | >512 | >512 | 256 | 512 | 512 | >512 |
| 1 | Olabogan | GRLIKFIKKAWRKFIEKFAK | 65 | 32 | 32 | 64 | 64 | 32 | 32 | 32 | 32 | 64 | 128 |
| 2 | Ratigan | GRILKFIKKAWRKIHLKFDK | >512 | 64 | 64 | 32 | 32 | 4 | 4 | 8 | 8 | 32 | 64 |
| 3 | Rudyxin | GRLLKFIKKAQRKFVEKFAK | >512 | 512 | >512 | 128 | 128 | 4 | 8 | 8 | 16 | 32 | 64 |
| 4 | Armaganan | GQLNKFIKKAWRKFFEKFAK | >512 | 64 | 128 | 8 | 8 | 8 | 8 | 4 | 4 | 32 | 128 |
| 5 | Sophieganan | GWLNKIIKKAWRKFHEIFSK | 167 | 4 | 4 | 4 | 8 | 8 | 8 | 2 | 2 | 32 | 32 |
| 6 | Piomiren | GRLNKFIKKAWRKFVLKFAK | >512 | 32 | 64 | 8 | 16 | 8 | 8 | 4 | 4 | 16 | 32 |
| | | Highly active | | 3 | 3 | 4 | 4 | 6 | 6 | 6 | 6 | 5 | 2 |
| | | Improved | | 5 | 5 | 6 | 6 | 6 | 6 | 6 | 6 | 6 | 6 |
| 0 | *Syphaxin* | GVLDILKGAAKDLAGHVATKVINKI | >512 | >512 | >512 | 512 | 512 | >512 | >512 | 128 | 256 | >512 | >512 |
| 1 | *Syphaxin-TI4* | GVLELLKGAAKDLAGHVATKVLKKI | >512 | >512 | >512 | 256 | 256 | 512 | 512 | 64 | 128 | >512 | >512 |
| 2 | Killixin | GVLDNLKSAAKDLAGHLATKVIKKI | 76 | 8 | 8 | 32 | 32 | 32 | 32 | 16 | 16 | 64 | 128 |
| 3 | Syphaxin-TII6 | GVLEILKGAAKDLAGHVATKVIKKI | >512 | >512 | >512 | 256 | 512 | 512 | 512 | 64 | 64 | >512 | >512 |
| 4 | Syphaxin-TII7 | GVLEILKSAAKDLAGHVATKVIKKI | >512 | 64 | 64 | 256 | 512 | 512 | 512 | 64 | 64 | 512 | >512 |
| 5 | Syphaxin-TII8 | GVIDHLKGGAKDLAGHVATKVIKKI | >512 | >512 | >512 | >512 | >512 | >512 | >512 | >512 | >512 | >512 | >512 |
| 6 | Syphaxin-TIII5 | GVIDFLKGAAKDLAGHVATKVIKKL | >512 | >512 | >512 | 256 | 512 | 512 | 512 | 64 | 64 | 512 | >512 |
| | | Highly active | | 1 | 1 | 1 | 1 | 1 | 1 | 1 | 1 | 0 | 0 |
| | | Improved | | 1 | 1 | 5 | 1 | 5 | 5 | 5 | 5 | 3 | 1 |
| 0 | *OP-145* | IGKEFKRIVERIKRFLRELVRPLR | >512 | 128 | 128 | 128 | 128 | 128 | 256 | 128 | 32 | 256 | >512 |
| 1 | Varsavian | IGKLFKRIHERIKRFLRVFLRRLR | 512 | 32 | 32 | 64 | 64 | 64 | 128 | 32 | 32 | 128 | 256 |
| 2 | OP-145-TI3 | IGKLFKRIHLRIKRFLRELVRQLR | 512 | >512 | >512 | 256 | 256 | 256 | 512 | 64 | 64 | >512 | >512 |
| 3 | Papan | IGKLFKRIVERIRRFVRSFLRTLR | 310 | 32 | 32 | 64 | 128 | 32 | 64 | 32 | 32 | 128 | 128 |
| 4 | OP-145-TII4 | IGKLFKRIVERIKRFLRVLLRILR | 166 | 128 | 256 | 256 | 256 | 256 | 256 | 256 | 256 | 256 | 256 |
| 5 | OP-145-TII5 | IGKLFKRIHERIKRFLRSFLRILR | >512 | 64 | 64 | 64 | 64 | 64 | 64 | 64 | 64 | 128 | 128 |
| 6 | OP-145-TIII4 | IGKEFKRIHERIKRFLRELLRHLR | >512 | 128 | 256 | 64 | 128 | 64 | 64 | 64 | 64 | 128 | 256 |
| | | Highly active | | 2 | 2 | 0 | 0 | 1 | 0 | 2 | 2 | 0 | 0 |
| | | Improved | | 3 | 3 | 4 | 2 | 5 | 4 | 0 | 0 | 4 | 5 |
| 0 | *Omiganan* | ILRWPWWPWRRK | >512 | 64 | 64 | 32 | 64 | 32 | 32 | 32 | 32 | 128 | 256 |
| 1 | Alacemycin | ILRWIWKIWRRW | >512 | 8 | 16 | 32 | 32 | 16 | 16 | 8 | 16 | 32 | 64 |
| 2 | Suselkan | GLRWIWKIWRRL | 95 | 4 | 4 | 8 | 16 | 4 | 8 | 4 | 4 | 16 | 32 |
| 3 | Omiganan-TII10 | ILRWPWRIWRR | >512 | 64 | 128 | 64 | 128 | 64 | 64 | 256 | 256 | 128 | 256 |
| 4 | Centaxin | LLRWIWRPWRR | >512 | 128 | 64 | 32 | 64 | 32 | 64 | 256 | 256 | 64 | 128 |
| 5 | Omiganan-TIII2 | ILRKPWRIWRR | >512 | 512 | >512 | 64 | 512 | 64 | 128 | >512 | >512 | 512 | >512 |
| 6 | Gedanan | FLRRFWRIWRR | 512 | 8 | 16 | 32 | 32 | 32 | 32 | 32 | 64 | 16 | 32 |
| | | Highly active | | 4 | 3 | 4 | 3 | 4 | 3 | 3 | 2 | 3 | 2 |
| | | Improved | | 4 | 3 | 1 | 3 | 2 | 2 | 3 | 3 | 4 | 4 |
| 0 | *Pexiganan* | GIGKFLKKAKKFGKAFVKILKK | >512 | 32 | 32 | 32 | 32 | 32 | 8 | 2 | 2 | 8 | 32 |
| 1 | Hydraganan-1 | GVGKKLWFALKKPLGLVKFFKLL | >512 | 8 | 8 | 8 | 8 | 16 | 16 | 4 | 4 | 32 | 64 |
| 2 | Hydraganan-2 | GIGKFLKFALKKGLGLVLKFKL | >512 | 32 | 32 | 128 | 128 | 128 | 128 | 32 | 32 | 128 | 256 |

**Table 2 (continued) | Experimentally validated analogues of GQ20, Syphaxin, OP-145, and Omiganan obtained through analogue generation with additional targeted preselection**

| ID | Name | Sequence | HC50 | S. aureus ATCC 33591 | | E. coli ATCC 25922 | | E. coli ATCC 43827 | | A. baumannii ATCC BAA 1605 | | P. aeruginosa ATCC 9027 | |
|---|---|---|---|---|---|---|---|---|---|---|---|---|---|
| | | | | MIC | MBC | MIC | MBC | MIC | MBC | MIC | MBC | MIC | MBC |
| | **Highly active** | | | 2 | 2 | 1 | 1 | 1 | 1 | 2 | 2 | 1 | 0 |
| | **Improved** | | | 1 | 1 | 1 | 1 | 0 | 0 | 0 | 0 | 0 | 0 |

Each row corresponds to a single peptide. MIC (μg/mL) values (c) and MBC (μg/mL) were determined against reference strains of microorganisms: *S. aureus* ATCC 33591, *E. coli* ATCC 25922, *E. coli* ATCC 43827, *A. baumannii* ATCC BAA 1605, *P. aeruginosa* ATCC 9027. HC50 was measured for sheep erythrocytes (μg/mL). All measurements were performed in triplicate. The row "Highly active" summarises the number of peptides (out of 6 candidate analogues) for each prototype that are highly active (MIC ≤ 32 μg/mL) against a given strain. The row "Improved" contains the number of peptides (out of 6 candidate analogues) that have improved activity against given strain with respect to the prototype. Values of HC50 that are greater than or equal 512 imply low toxicity and are printed in bold. Values of MIC or MBC that are less than or equal 128 indicate antimicrobial activity and are printed in bold. Values of MIC or MBC that are less than or equal 32 indicate a high antimicrobial activity and are printed with the underline. Additional measurements for Hydraganan-1 and Hydraganan-2 were also performed and placed at the end of the table.

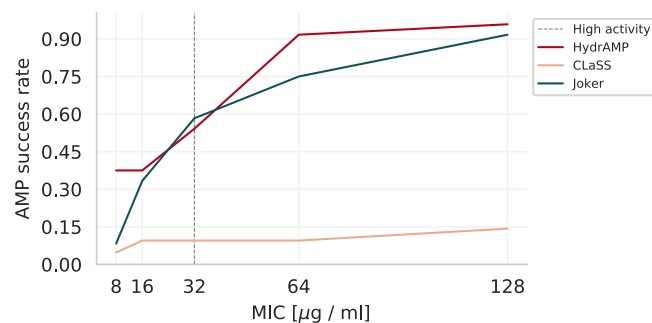

**Fig. 7 | AMP success rates depending on activity threshold.** AMP success rate (y-axis) for different MIC thresholds (MIC measured in units [μg / mL]; x-axis), for different methods: HydrAMP (red), CLaSS (beige), Joker (dark green). Source data are provided as a Source Data file.

Hydraganan-2 against two of them. For *S. aureus* and *E. coli* ATCC 25922, Hydraganan-1 showed MIC of 8 μg/mL, while for *A. baumannii* it obtained MIC = 4 μg/mL. Although the exact MIC measurements for the two *E. coli* strains deviated from the ones obtained in the first experimental round (compare Table 1), the relative improvement of activity of Hydraganan-1 over Pexiganan, as well as its higher activity as compared to Hydraganan-2 was confirmed in both experimental rounds.

Since HydrAMP was trained on peptides active against *E. coli* strains, the experimental results on this species are the most important for validation of its generative power. For *E. coli* ATCC 25922, the most prevalent strain in the GRAMPA database[49], 9 out of 24 tested peptides were validated as highly active (MIC≤ 32 μg/mL; Table 2), corresponding to a positive validation rate of 38%, larger than the positive validation rate of 29% obtained in the previous experimental round without additional preselection of the most promising candidates (compare Table 1). For the more permissive activity threshold of MIC ≤ 128 μg/mL, the positive validation rate in this experimental round was much higher (71%). In total, 16 out of 24 (67%) obtained analogues had an improved activity against *E. coli* ATCC 25922 compared to their prototypes. The improved analogues of Omiganan and Pexiganan are of particular importance, as these prototypes have very good starting MIC against *E. coli* ATCC 25922 of 32 μg/mL and 16 μg/mL, respectively. The validation results for *E. coli* should be largely similar between the two different strains (+/- one dilution in the dilution series). As expected, this similarity was observed for most of the peptides (Table 2). In summary, these results validate excellent generative performance of HydrAMP, and demonstrate that the proposed preselection procedure is able to determine the most promising candidates, decreasing the cost of the experimental effort.

The comparison of activity of the peptides against *E. coli* to two additional gram negative bacteria revealed interesting patterns. Generally, activity against *A. baumannii* tended to be higher than against *P. aeruginosa* (21 out of 24 peptides had lower or equal MIC against *A. baumannii*). At the same time, activity of peptides against *E. coli* was more similar to their activity against *A. baumannii* than to *P. aeruginosa*. Notably, among AMPs identified in this experimental round, we obtained four peptides with a remarkable activity profile against *A. baumannii*, namely Armaganan, Piomiren, Sophieganan and Suselkan, all with MIC ≤ 4 μg/mL.

We also compared the activity of the newly generated peptides between the gram-negative and gram-positive (*S. aureus*) species. The relative activity of peptides against *S. aureus* versus the remaining bacterial strains varied from case to case and there was no clear tendency. In particular, we cannot conclude whether peptides generated by HydrAMP are more effective/active towards gram negative or gram-positive bacteria. For example, the new AMP called Rudyxin, was inactive against *S. aureus* (MIC = 512 μg/mL), and showed activity

against *E. coli* (ATCC 25922 MIC = 128 *μg*/mL, ATCC 43827 MIC = 4 *μg*/mL). In contrast, the AMP called Hydraganan-2, showed high activity against *S. aureus* (ATCC 33491 MIC = 32 *μg*/mL), but was less active against *E. coli* (ATCC 29522 MIC = 128 *μg*/mL). Six peptides showed high activity (MIC ≤ 32 *μg*/mL) both against gram-negative *E. coli* and gram-positive *S. aureus*, namely Alacemycin, Centaxin, Gedanan, Killixin, Sophieganan and Suselkan.

Additionally, we tested the toxicity of all of the candidates measured as peptide concentration causing 50% hemolysis (HC50; see Methods). As much as 18 out of 24 considered analogues (75%) were proven completely non-toxic (HC50 ≥ 512 *μg*/mL). Moreover, 6 tested peptides showed both high activity against *E. coli* ATCC 25922 and low toxicity. Hemolysis assays are standard practice in initial toxicity testing[50-55]. Additionally, some studies indicate that erythrocytes are more sensitive to the action of cationic antimicrobial peptides than cell lines, such as HaCaT or HeLa, which are often used in cytotoxicity assays[56]. Thus, the observed low toxicity in the hemolysis assay may also be indicative of low cytotoxicity.

Remarkably, all six generated analogues of the negative prototype GQ20 showed high activity against at least three out of five tested bacterial strains, as well as low toxicity (Table 2). Notably, these novel peptides bear no resemblance to any of the existing AMPs, as they were generated from a negative prototype, and as such are of particular interest. One of the analogues of GQ20, a peptide named Ratigan, tested active against four bacterial strains, obtaining MIC = 32 *μg*/mL against *E. coli* ATCC 25922, MIC = 4 *μg*/mL against *E. coli* ATCC 43827, as well as MIC = 8 *μg*/mL against *A. baumannii*, and MIC = 32 *μg*/mL against *P. aeruginosa*. Another GQ20 analogue, named Armaganan, was active against the same four bacterial strains, with MIC = 8 *μg*/mL for both *E. coli* strains, MIC = 4 *μg*/mL for *A. baumannii*, and MIC = 32 *μg*/mL against *P. aeruginosa*. Both these interesting peptides are non-toxic, with HC50 ≥ 512 *μg*/mL. An analogue of GQ20, named Sophieganan, had only a slightly higher toxicity (HC50 = 248 *μg*/mL) and proved highly active against all five bacterial strains, with MIC = 4 *μg*/mL against *S. aureus*, 8 *μg*/mL for the two *E. coli* strains, MIC = 4 *μg*/mL for *A. baumannii*, and MIC = 16 *μg*/mL for *P. aeruginosa*.

Taken together, the experimental validation results confirmed the capability of our framework in the task of generating novel, potent peptides, which are active against a range of aggressive and resistant bacteria, and which could potentially be used in the fight against the antimicrobial resistance crisis.

### Molecular structures of the newly discovered AMPs

Finally, we investigated molecular structures for all 180 peptides that were simulated in the prelection phase (Fig. 8a). As expected, MD simulations revealed that most peptides remained *α*-helical, with minor to moderate unstructured regions at chains termini. Only in few cases, unfolding involved more than 50% of residues. Notably, there was no straightforward correlation between the degree of helix stability and the depth of membrane penetration (Fig. 8b): some fully helical peptides were found to only loosely associate with the membrane ($S < 0.25$), while some unstructured ones were found to insert deeply ($S > 0.75$). We note that the actual antimicrobial action typically relies on the achievement of critical peptides concentration, which promotes their mutual interactions and facilitates further translocation through the membrane core. Accordingly, the extent of secondary structure present at functional conditions may be different than observed in our simulations. Still, we consider them as more reliable and closer to the truth than simulations or predictions that do not account for the presence of the bacterial membrane.

The molecular structures for the subset of peptides that were tested in the second experimental round mostly obtained helical structures (Supplementary Figs. 6–9). We observed *α*-helices, *α*-helices with a kink, as well as incomplete *α*-helices.

Intrigued by the reported success in protein structure prediction by AlphaFold2[57], we additionally investigated its performance in the context of our considered sequences. In all cases we obtained uniformly *α*-helical geometries (Fig. 8a), which indicates a tendency to produce overly stable secondary structure, at least in comparison to the simulation model. However, this result may stem from the fact that, unlike the simulation-based structures, the AlphaFold2 predictions are not aware of the presence of the microbial membrane. Moreover, AlphaFold2 was not trained to show structure differences in proteins with small changes in amino acids composition and many analogues of peptides differ in barely couple of monomers. It is important to note that AlphaFold2 was not designed to work with short sequences as its training inputs are usually considerably longer than average length of an AMP. Thus, only the fully atomistic MD simulations can be reliably used to study the conformation of generated analogues in membrane environment.

## Discussion

In this work we have proposed HydrAMP, a generative model for antimicrobial peptides discovery. It leverages a conditional variational autoencoder to offer two functionalities: generating analogues of existing peptides with specified antimicrobial properties (analogue generation) and generating peptides de novo (unconstrained generation). This is enabled by a continuous peptide representation of reduced dimensionality with disentangled antimicrobial conditions $c_{AMP}$, $c_{MIC}$. Additionally, this representation is directly optimized to not only properly represent the known peptides but also to efficiently generate new candidates. To facilitate the usage of HydrAMP model, we developed a web service available freely at https://hydramp.mimuw.edu.pl/.

We have evaluated the model's ability to improve existing antimicrobial peptides by producing antimicrobial analogues of known AMPs: Pexiganan, Omiganan, Syphaxin, and OP-145. All of the obtained analogues were comprehensively evaluated both in terms of activity as well as structure. First, we measured both MIC and MBC for a set of five strains, including both gram-positive and gram-negative species, as well as resistant strains. Second, we also measured the toxicity of all the analogues, given as HC50 for mammalian erythrocytes. Finally, we simulated the structure and behaviour of the analogues in contact with the bacterial membrane.

HydrAMP bears several novelties and advantages in comparison to existing approaches. First, in contrast to previous VAE/WAE-based approaches[27,29,30], HydrAMP was trained specifically for the task of analogue generation. In particular, we trained HydrAMP both to generate positive (active) and negative (inactive) analogues. As we are predominantly interested in generating active prototypes, the latter task can be viewed as a form of model regularization, making the model more generalizable. Additionally, HydrAMP is the first generative model trained and used to identify active analogues of non-AMP prototypes. In this way, we increase the diversity pool of the generated analogues. Third, contrary to most currently applied methods, we leverage the output distribution of the Decoder by sampling multiple times from this distribution to obtain potential candidates. To mimic such sampling during the training, we applied the Gumbel Softmax approximation (Methods). In contrast, PepCVAE and Basic, as well as other VAE based methods[27,29], take only the *maximum a posteriori* from the decoder distribution. Leveraging the Gumbel Softmax approximation[58] enabled a continuous approximation of sampling in the discrete space of amino acid sequences and thus a direct optimization of peptides generated by the model. Before, such optimizations required a complex multi-stage training[29]. Finally, HydrAMP is the only model, which controls in a parametrized way its creativity understood as the number of modifications introduced to the query peptide.

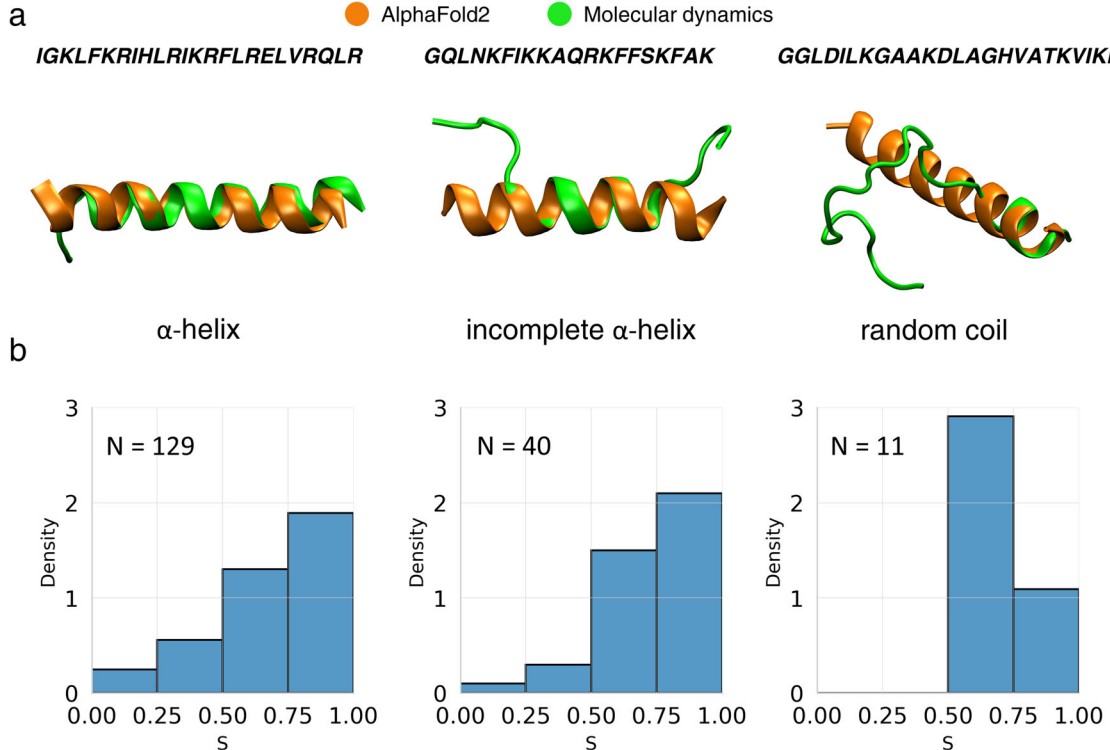

**Fig. 8 | Peptide structures according to MD simulations and AlphaFold2 predictions. a** Left to right: representative structures for the last 100 ns of MD runs, for peptides with average helical content in [0.75, 1], [0.5, 0.75), (0,0.5) range, respectively (green), with superimposed AlphaFold2 predicted geometries (orange); **b** $S$ distributions in corresponding peptide groups. Source data are provided as a Source Data file.

Although HydrAMP conveys advancement over previous approaches, it still could be extended in several ways. First, HydrAMP could be enriched with an attention mechanism[59]. The architecture of the model allows easy replacement of both the Encoder and the Decoder with transformer modules[59]. It is not clear, however, that such an extension of the model would improve its performance, as initial evidence suggested that attention-based models might work worse for peptide modelling than recurrent models[60]. In HydrAMP we leveraged the AMP classifier which was trained on a negative dataset consisting of UniProt peptides that do not show an AMP flag (see Methods). We also used the MIC classifier, for which the negative dataset consisted of peptides with low MIC against *E. coli*. Instead, it would be preferable to use endogenous *E. coli* peptides that are unlikely to affect *E. coli* cells. However, this approach would be not feasible, as HydrAMP is a deep learning method, and as such requires a larger number of training samples. Additionally, we considered amino acid sequence only. Instead, we could leverage structure prediction models such as AlphaFold2[57], and extend HydrAMP training procedure to directly optimize the alpha-helix propensity and other structural features of AMPs. However, our evaluation of the structures predicted by Alpha-Fold2, revealed deviations from the structures obtained using MD in the presence of the bacterial membrane. In this work, the loss term responsible for the reconstruction objective treats each of the amino acids in the sequence as independent. However, the probability of each amino acid being present is dependent on proceeding and following amino acids. The local context can also affect the antimicrobial activity of a given peptide, as shown in[61]. Modelling of such dependencies could be achieved by introducing autoregression or attention mechanism to the Decoder, at the cost of increasing the complexity of the model, which might decrease its generative abilities[62].

Finally, the applicability of the model could be also extended. HydrAMP does not consider the host toxicity of the peptides. Successfully optimizing both activity and host toxicity within the same model would provide a powerful tool for AMP design. Addressing the toxicity is an urgent matter as only highly selective peptides can be used in treatment of human subjects. Additionally, we trained the model to target only *E. coli* strains. Depending on the data availability, it could be retrained to target other strains, either gram positive or multi-drug resistant strains.

Antimicrobial resistance problem remains a serious threat that needs dire actions. The high success rate in experimental validation, the high measured activity against a wide range of bacteria, including multi-antibiotic resistant strains, and non-toxicity of the peptides generated by HydrAMP, confirmed the applicability of our framework for generating highly promising antimicrobial peptides against clinically relevant bacteria. In summary, HydrAMP emerges as a promising tool for designing novel antibiotics.

## Methods

### Data collection
The training data for the model is a curated data set of a total of 247 506 peptide sequences collected from three sources: MIC data collected by[12], AMP data from known AMP databases, and UniProt data[63] (Fig. 1a). All selected training sequences have length of at most 25 amino acids and contain only standard amino acids. Sequences shorter than 25 amino acids were padded using additional padding character. The threshold of 25 amino acids for the length of peptide sequences is chosen for two reasons: availability of training data and the ability of synthesizing such peptides in the lab. AMPs up to 25 amino acids long constitute the majority of peptides with length up to 100 amino acids (11131/19883; 56%; Supplementary Fig. 10). This amounts to 11131/20313 (55%) of all available AMP sequences. At the same time the threshold of 25 amino acids is a reasonable synthesizability cut-off. Based on our experience, sequences up to this length usually yield good purity (>60%) of raw products.

**MIC data.** The MIC data set consists of peptides with experimentally proven antimicrobial activity and measured MIC values downloaded from GRAMPA database[12]. Of 8049 entries only 4546 of them are unique sequences. We select those peptides which were experimentally tested for activity against *E. coli* strains. MIC values for peptides with multiple measurements are averaged resulting in 4546 entries. Only 3444 peptides have the required length of 25 amino acids. Peptides with $MIC \leq 10^{1.5} \simeq 32\,\mu g/mL$ are labeled as active (positive), while peptides with $MIC > 10^{1.5} \simeq 32$ are labelled as inactive (negative). The MIC data set contains 2126 positives and 1318 negatives.

**AMP data.** The AMP data consists of positive and negative data set. Positive examples are sequences from manually curated AMP databases. We combine experimentally validated peptides from dbAMP[64], AMP Scanner[8], and DRAMP[65] databases. Duplicate sequences are removed. Negative examples are assumed to be biologically inactive and are obtained manually using UniProt search filters, requiring subcellular location: cytoplasm, and excluding the following properties: antimicrobial, antibiotic, antiviral, antifungal, effector, excreted. To increase data set diversity, negative sequences sharing $\geq 40\%$ sequence identity are removed, and replaced with the representative sequences of the clusters using CD-HIT[66,67]. The resulting set of representative sequences contains only sequences that are shorter than or 25 amino acids long, thus negative sequences are randomly cropped to match the positive data set length distribution in order to avoid length bias. The positive and negative data sets are of equal size of 11,131 sequences each.

**UniProt data.** To extend the biological diversity in the training process we downloaded sequences of unknown antimicrobial properties of the desired length from UniProt database. This gave 225 244 additional sequences with no duplicates.

## AMP and MIC classifiers learning

In order to predict the properties of generated peptides, we trained a pair of classifiers that, for a given peptide $p$, predict its probabilities of being antimicrobial and active. The reason why we have decided to use two separate classifiers is that we make a separation between AMPs in general and AMPs active against *E. coli*. In this work, we understand AMPs as all the peptides that are deposited in databases such as DBAASP or CAMP. In turn, we use the term "active" referring to peptides which have $MIC \leq 10^{1.5} \simeq 32\,\mu g/mL$ against *E. coli* strains, as this is the most prevalent species in GRAMPA database[12]. The conservative threshold of 32 μg/mL was deliberately chosen so that the model focused on the highly active peptides in the training set. The distinction between an AMP and active peptide is important as there exist multiple AMPs, which are considered AMP and are not active against *E. coli*. This motivates the use of two classifiers, where AMP classifier is trained on all the available AMP sequences, and MIC classifier is trained on a smaller subset of sequences with MIC measurements against *E. coli* available. Incorporating the predictions of both these classifiers helps HydrAMP to generate peptides which are both AMP and active against *E. coli*. For the prediction of a given peptide being antimicrobial (AMP), we implemented and trained the model from AMP Scanner[8], using AMP data. We refer to this model as $M_{AMP}$. For the prediction of peptide activity, we trained a model using MIC data, denoted $M_{MIC}$, with a custom architecture. The model consists of six layers. The first layer is an one-hot encoded input in a form of a sequence of amino acids. The second layer is an 128-dimensional embedding layer of each of the amino acids. Next, we use the LSTM layer of 64 units[68]. Downsampling is performed by 1-dimensional pooling with kernel size of 5 and stride of 5. Another LSTM layer of 100 units is used where only the last output is returned. The final output is a fully connected layer with a sigmoid activation.

For a given peptide $p$, the $M_{AMP}$ returns the probability of $p$ being antimicrobial, denoted $\mathbb{P}_{M_{AMP}}(p)$. The $M_{MIC}$ returns the probability of $p$

being active, denoted $\mathbb{P}_{M_{MIC}}(p)$. From now on we assume that these probabilities are obtained only using $M_{AMP}$ and $M_{MIC}$ classifiers, and we refer to them jointly as Classifier. Both models are implemented in Keras[69].

Cross-validation results indicate highly accurate classification results for both $M_{AMP}$ and $M_{MIC}$ (Supplementary Table 1).

## HydrAMP model

HydrAMP is an extended conditional VAE (cVAE) model. Its main generative part is a Decoder model that given a *latent variable* $z \in \mathbb{R}^{latent}$ and a pair of conditions $\mathbf{c} = (c_{AMP}, c_{MIC})$ produces a distribution over peptides $Dec(z, \mathbf{c})$. We refer to the likelihood of a peptide $p$ w.r.t. to this distribution as $\mathbb{P}_{Dec(z,\mathbf{c})}(p)$. The latent variable is assumed to follow the *latent prior* distribution $P_z \sim \mathcal{N}(0, I)$.

The Decoder is trained so that the peptides sampled from $Dec(z, \mathbf{c})$ follow a given pair of conditions $\mathbf{c}$ and resemble valid peptides from the training dataset. As optimizing the Decoder directly is not feasible, we introduced an additional Encoder model trained to provide a variational approximation $q(z|p)$ of the posterior distribution $\mathbb{P}(z|p)$. The variational posterior approximation for a peptide $p$ is set to be a normal distribution $\mathcal{N}\left(\mu_{q(\cdot|p)}, diag(\sigma_{q(\cdot|p)})\right)$, where $\mu_{q(\cdot|p)} \in \mathbb{R}^{latent}$ and $\sigma_{q(\cdot|p)} \in \mathbb{R}_+^{latent}$. Both the Decoder and Encoder are modeled as neural networks. For the detailed architecture see Supplementary Methods 4.

The HydrAMP model is optimized using three objectives: reconstruction, analogue (see Supplementary Methods 5.1), and unconstrained (see Supplementary Methods 5.2). The reconstruction objective aims at teaching the Decoder how to generate valid peptide structures by reconstructing known peptides. The analogue and unconstrained objectives mimic the process of the analogue and unconstrained generation during training. Besides that, we also applied a two-fold regularization to the HydrAMP model: the Jacobian disentanglement regularization to encourage disentanglement between latent variable $z$ and a pair of conditions $\mathbf{c}$, and latent reconstruction regularization for better latent variable preservation properties.

In the formulas below we use the following notation. Denote a constant $c \in [0, 1]$, $p$ a peptide and $P_M(p)$ its probability in a classifier model $M$, with $M \in \{M_{AMP}, M_{MIC}\}$. Let

$$H_M(c, p) = \log\left(\mathbb{P}_M(p)^c(1 - \mathbb{P}_M(p))^{1-c}\right), \qquad (1)$$

be the cross-entropy between the *Bernoulli*(c) and *Bernoulli*($\mathbb{P}_M(p)$). We define

$$H_\Sigma(\mathbf{c} = (c_{AMP}, c_{MIC}), p) = \Sigma_{cond \in \{AMP, MIC\}} H_{M_{cond}}(c_{cond}, p), \qquad (2)$$

where $\mathbf{c} = (c_{AMP}, c_{MIC})$ is the pair of conditions, as the sum of the cross entropies for the two different conditions $c_{AMP}, c_{MIC}$ and their probabilities $\mathbb{P}_{M_{MIC}}(p)$ and $\mathbb{P}_{M_{AMP}}(p)$, respectively.

**Reconstruction objective.** The reconstruction objective forces the model to capture the structure of valid peptides collected from available databases. We achieve this by training the Decoder and Encoder maximizing the conditional evidence lower bound (ELBO) introduced in[70]. For each peptide $p \sim \mathbb{P}_\chi$, we compute its pair of conditions $\mathbf{c}_p = (\mathbb{P}_{AMP}(p), \mathbb{P}_{MIC}(p))$ using the Classifier. Next, we maximize a conditional ELBO given by:

$$ELBO_{rec}^\beta = \mathbb{E}_{z \sim q(z|p)}(\log \mathbb{P}_{Dec(z, \mathbf{c}_p)}(p) + \qquad (3)$$

$$\lambda_{rec}^{class} \mathbb{E}_{p' \sim Dec(z, \mathbf{c}_p)} H_\Sigma\left(\mathbf{c}_p, p'\right)) - \qquad (4)$$

$$\beta \cdot KL(q(z|p) \parallel P_z), \qquad (5)$$

where (3) is the expected log likelihood of reconstruction of the initial peptide $p$, (4) is the expected log likelihood of recovering the initial peptide pair of conditions $\mathbf{c}_p$ with parameter $\lambda_{rec}^{class} > 0$ and (5) is a $\beta$-VAE regularization term[71] with parameter $\beta > 0$ decaying in the process of training. Parameters $\beta$ and $\lambda_{rec}^{class}$ control the trade-off between reconstruction of the original peptide, satisfying the peptide condition and keeping the posterior approximation $q(z|p)$ close to the prior. The expectations w.r.t. $q(z|p)$ are obtained using a reparametrization trick[72] and are approximated using a single sample from $q(z|p)$.

**Jacobian disentanglement regularization.** For every generation process used by the HydrAMP model, a newly generated peptide should have properties provided by a pair of conditions $\mathbf{c}$. To measure that, let us define average condition reconstruction functions for being AMP ($ACR_{AMP}$), and being MIC ($ACR_{MIC}$) of the $Dec$ distribution as:

$$ACR_{AMP}(z, c = (c_{AMP}, c_{MIC})) = \mathbb{E}_{p \sim Dec(z,c)} \mathbb{P}_{AMP}(p) \quad (6)$$

and

$$ACR_{MIC}(z, c = (c_{AMP}, c_{MIC})) = \mathbb{E}_{p \sim Dec(z,c)} \mathbb{P}_{MIC}(p). \quad (7)$$

Now let $ACR = (ACR_{AMP}, ACR_{MIC})$. It measures the expected likelihood of the peptides sampled from the distribution $Dec$ modeled by the Decoder, for a given $z$ and pair of conditions $\mathbf{c}$, actually satisfying $\mathbf{c}$. Ideally $ACR(z, \mathbf{c}) = \mathbf{c}$, which means that on average peptides sampled from $Dec(z, \mathbf{c})$ have properties defined by the pair of conditions $\mathbf{c}$. This means that in an ideal scenario the $ACR$ is constant w.r.t. $z$ and in case when $ACR$ is differentiable w.r.t. to $z$, the following condition holds:

$$\frac{\partial ACR}{\partial z} \equiv \mathbf{0}, \quad (8)$$

where $\mathbf{0}$ is an all zero matrix. In that case $z$ and $\mathbf{c}$ are disentangled w.r.t. to $ACR$, because any change of $z$ does not affect the expected pair of conditions of newly generated candidates. In order to impose this property we introduce the following Jacobian disentanglement regularization function:

$$JDR^{Dec}(z, \mathbf{c}) = \sum_{cond \in \{AMP, MIC\}} \sum_{j=1}^{latent} Huber\left(\frac{\partial ACR_{cond}(z, \mathbf{c})}{\partial z_j}\right), \quad (9)$$

where $Huber : \mathbb{R} \to \mathbb{R}$ (sometimes also referred as a smooth $L1$) is given by

$$Huber(x) = \min(|x|, x^2). \quad (10)$$

We use the Huber function as it is less prone to be affected by outlier examples.

Accordingly, we extended reconstruction objective with the following term:

$$JDR_{rec} = \mathbb{E}_{z \sim q(z|p)} JDR^{Dec}(z, \mathbf{c}_p), \quad (11)$$

where $p$ is a peptide being reconstructed and $\mathbf{c}_p$ is its pair of conditions. We approximate this expectation with a single sample from $q(z|p)$ using the reparametrization trick.

Analogue and unconstrained objectives were also extended with the Jacobian disentanglement regularization term. See Supplementary Methods 6.1 for the details.

**Latent reconstruction regularization.** In the cVAE framework, the Decoder plays a role similar to the inverse function of the Encoder.

Indeed, the Decoder aims to reconstruct the peptide fed to the Encoder that is sampled from a posterior distribution generated by the Encoder. To further impose that relation, similarly to[29], we introduced an additional latent reconstruction regularization objective. Consider a peptide $p$ and its posterior mean $\mu_{q(z|p)}$ given by the Encoder. Peptide $p'$ returned by the Decoder for a point sampled from that posterior can be given as input to the Encoder and will obtain its posterior mean $\mu_{q(z|p')}$. The latent reconstruction regularization objective enforces the two posterior means to be similar. To this end, we minimize the following expectation for the reconstruction objective:

$$LRR_{rec} = \mathbb{E}_{z \sim q(z|p)} \mathbb{E}_{p' \sim Dec(z, \mathbf{c}_p)} \left\| \mu_{q(\cdot|p)} - \mu_{q(\cdot|p')} \right\|_2^2, \quad (12)$$

where $p$ is a reconstructed peptide and $\mathbf{c}_p$ is its pair of conditions. Using a reparametrization trick, we approximate this expectation with a single sample from $q(z|p)$ and the expectation w.r.t. $Dec$ is approximated using a Gumbel Softmax[58].

When the expectation above is low, the Encoder preserves the latent code of the average peptide sampled from the Decoder. This property is essential especially for the analogue generation, as we assume the similarity between the analogue and original prototype because we sample both from precisely the single posterior distribution over latent codes. If the Encoder preserves the latent code of a generated analogue, then its similarity to prototype is expected for continuous Decoder models trained in the VAE framework.

Analogue and unconstrained objectives were also extended with the latent reconstruction regularization terms. See Supplementary Methods 6.2 for the details.

### Basic and PepCVAE models
We compared HydrAMP with two other models: *Basic* and *PepCVAE*[29]. We used our own implementation of the PepCVAE model, as its code was not made publicly available by the authors.

*Basic* model corresponds to the standard cVAE model (see[70]), and is trained in the same manner as HydrAMP, but is optimized only for reconstruction objective and uses only the latent reconstruction regularization. PepCVAE model was trained to optimize the same objectives as Basic, but was additionally optimized for the unconstrained objective. Both PepCVAE and Basic models lack optimization of analogue objective and Jacobian disentanglement regularization, which are incorporated in HydrAMP.

Since the Basic, PepCVAE and HydrAMP are increasingly complex, such a selection of models for comparison effectively implements an ablation study.

**Training procedure.** We trained all models (HydrAMP, Basic, and PepCVAE) using ADAM[73] optimizer. The batch size was equal to 384, and each batch consisted of 128 peptides from AMP, 128 from MIC data, and 128 from UniProt data. We trained every model for 40 epochs. Reconstruction and analogue objectives were optimized using all peptides in each batch. Additionally, in each iteration, the unconstrained objective was optimized using 128 samples from $P_z$. The Gumbel temperature used in Gumbel Softmax[58], $t$, was scheduled using exponential decay from 2.0 to 0.1 for 24 epochs and then kept stable at 0.1. The $\beta$ parameter was increased from $10^{-4}$ to $10^{-2}$ via exponential annealing.

For the loss function, evaluation metrics and model selection details for training of HydrAMP, Basic and PepCVAE models see Supplementary Methods 5–7.

### Post-training prior refinement
Following successful results presented in[74], after the end of the training, we refined our $P_z$ prior distribution to better match an aggregated

posterior:

$$\mathbb{P}_z^{agg} = \mathbb{E}_{p \in \mathcal{X}} q(\cdot|p), \qquad (13)$$

which is an average of all posterior distributions of peptides from dataset $\mathcal{X}$. According to[75] this distribution is the latent prior distribution which maximizes likelihood of data from $\mathcal{X}$ when Encoder and Decoder models are fixed. However, this property makes it prone to over-fitting. Because of that we decided to use less complicated distribution to approximate the aggregated posterior. This new distribution $\hat{\mathbb{P}}_z^{agg}$ is set to be a normal distribution $\hat{\mathbb{P}}_z^{agg} = \mathcal{N}\left(\hat{\mu}, \hat{\Sigma}\right)$ optimized to maximize likelihood of the set of aggregated variational posterior means $\{\mu_{q(\cdot|p)}|p \in \mathcal{X}\}$. The distribution parameters $\hat{\mu}, \hat{\Sigma}$ are selected using a classical PCA algorithm[76].

## Generation modes of peptides
After the model is trained we use it to generate novel peptides. There are the two modes of this process: analogue generation and unconstrained generation.

## Analogue generation
For analogue generation, Supplementary Algorithms 1 and 2 are used for generation of active peptides similar to prototype peptide $p_{proto}$. In these algorithms we introduce a creativity parameter $\tau \in (1, +\infty)$. Analogues are sampled from a modified variational posterior $\mathcal{N}(\mu_{q(\cdot|p_{proto})}, \tau^2 \cdot \sigma_{q(\cdot|p_{proto})})$ with a covariance matrix rescaled by a factor of $\tau^2$. This means that the closer $\tau$ is to 1 - the more similar the sampled analogues are to $p_{proto}$. On the other hand - for $\tau > 1$ the sampling probability distribution has the same mean but greater variance than in case of posterior approximation what encourages generating peptides differing from $p_{proto}$ to a greater degree.

An analogue meets the improvement discovery criteria when it increased $\mathbb{P}_{M_{AMP}}(p)$ and $\mathbb{P}_{M_{MIC}}(p)$ with respect to the input peptide. An analogue meets the baseline discovery criteria with $\mathbb{P}_{M_{AMP}}(p) \geq 0.8$ and $\mathbb{P}_{M_{MIC}}(p) > 0.5$.

## Unconstrained generation
Supplementary Algorithm 5 is used for generation of active and antimicrobial peptides in an unconstrained manner where $z$ is sampled from a refined prior $\sim \hat{\mathbb{P}}_z^{agg}$. Supplementary Algorithm 5 refers to the positive mode. In the negative mode we sample peptides with conditions $(c^{AMP} = 0, c^{MIC} = 0)$, filter out peptides with $\mathbb{P}_{M_{AMP}}(p) > 0.2$, and select the peptide with the lowest $\mathbb{P}_{M_{MIC}}(p)$.

## Generation using additional models
Generation using Basic and PepCVAE is described in Supplementary Methods 1 and 2. The important difference between HydrAMP and these methods is that while all of them model the Decoder as a probability distribution, only HydrAMP samples the candidate peptides from that distribution. For Basic and PepCVAE, the returned peptides are computed as the *maximum a posteriori* samples from the Decoder, which significantly decreases the diversity of generated peptides.

## Biological filtering criteria
In general, the biological criteria serve as approximation of expert selection that takes into account peptide synthesizability. First, we filter out all of the known AMPs we collected in our AMP data set. Then, we exclude sequences in which in a window of 5 amino acids there were more than 3 positively charged amino acids (K, R). Finally, we remove sequences in which occur three hydrophobic amino acids in a row. We consider as hydrophobic following amino acids based on Eisenberg scale[77]: F, I, L, V, W, M, A.

In case of selection of peptides for experimental validation we use more stringent criteria. We remove sequences of known AMPs, and

sequences with accumulation of positive charge, as described above. Additionally, we remove any sequence in which three amino acids in a row are the same. We also exclude sequences containing cysteines (C).

## Computer simulations of peptide-membrane systems
A generic simulation box constructed with the CHARMM-GUI service[78] comprised a rectangular lipid bilayer patch consisting of 120 1-palmitoyl-2-oleoyl-sn-glycero-3-phosphoethanolamine (POPE) and 60 1-palmitoyl-2-oleoyl-sn-glycero-3-(phospho-rac-(1-glycerol)) (POPG) molecules, embedded in aqueous solvent with $K^+$ and $Cl^-$ ions, whose number was chosen to achieve 0.15 mol/l concentration and subsequently adjusted to neutralise the total system charge. Initial box dimension across the membrane was 11 nm. All peptide starting structures were modelled as regular $\alpha$-helices using the Discovery Studio Visualizer 2021 program (Dassault Systèmes, BIOVIA)[79] with standard protonation states of titratable residues, charged N-termini and amidated C-termini. CHARMM36 force field[80] was used for protein, lipids and ions, and rigid TIP3P model[81] was used for water. During final system assembly peptide structures were inserted into the aqueous compartment of the simulation box with their helical axes oriented parallel to the membrane plane and overlapping water molecules were removed. Simulations were carried out using the Gromacs program[82] with the default simulation set up implemented in CHARMM-GUI for the chosen force field combination and periodic boundary conditions[83]. The protocol included potential energy minimisation, six rounds of equilibration with stepwise removal of positional restraints for peptide and lipids, and production runs. The latter were conducted at ambient pressure and temperature of 310 K.

The stability of peptides $\alpha$-helical structure was assessed by the DSSP program[84]. The degree of peptide-membrane association for a simulation interval of interest was evaluated by $S$ parameter. It was obtained by calculating probability distributions of finding protein and lipid heavy atoms along the $Z$ axis perpendicular to membrane plane, $\rho_p(z)$, and $\rho_l(z)$, respectively, and defining $S = \int_0^{z_0} \rho_p(z)dz$, with $z = 0$ denoting membrane midplane, and $z_0$ chosen such that $\rho_l(z_0) = 0.5\max(\rho_l)$.

In the study of Hydraganan and inactive Pexiganan analogue, the peptides were placed 3.5 nm from the membrane surface, with random initial orientation along the helical axis. Two additional transformations by 120 and 240 degrees, respectively, were generated, resulting in three starting structures in either case, each simulated independently with production phase of 1 $\mu s$.

In the first round of the screening protocol (MD1), the peptides were placed 3.0 nm from lipid bilayer surface in such orientation along their helical axes as to maximize the total positive side chain charge exposed towards the negatively charged membrane. Following standard equilibration steps, the systems were simulated for 1 $\mu s$. In the second round of the screening protocol (MD2), the peptides were placed 0.3 nm from lipid bilayer surface in such orientation along their helical axes as to maximize the exposure of hydrophobic residues towards the membrane, with hydrophobicity score for each residue taken from the Kyte and Doolittle scale[85]. In addition to standard equilibration steps, a 50 ns run was included prior to production, in which the peptide centre of mass was held next to the centre of mass of phosphate atoms of the neighbouring membrane leaflet by a harmonic potential with 1000 kJ/nm² force constant. Subsequently, each system was subjected to 500 ns of unconstrained simulation.

## Wet-lab evaluation
**Synthesis.** The synthesis was carried out automatically on a microwave Liberty Blue™ Automated Microwave Peptide Synthesizer (CEM Corporation, Mathews, NC, USA), equipped with a fiber optic temperature probe and a gas cooling system. Peptides were synthesized by solid-phase Fmoc/tBu methodology, where polystyrene resin modified

by Fmoc-Rink Amide (Orpegen Peptide Chemicals GmbH, loading 1.0 mmol/g Heidelberg, Germany) linker were used as solid support. Synthesis scale of each peptide was 0.1 mmol. Before synthesis resin was swelled with DFM (*N,N*-dimethylformamide; POCH, Avantor, Gliwice, Poland) for 5 min at RT. Deprotection of the Fmoc group was performed with 20% (v/v) piperidine (Iris Biotech GmbH, Marktredwitz, Germany) solution in DMF. Couplings were performed with an equimolar mixture of Fmoc-AA-OH, DIC and Oxyma Pure in fivefold excess based on the resin. Each cycle consist of deprotection step, washing of the resin and subsequent coupling of Fmoc-protected amino acid. Deprotection was performed at two stages. First at 75 °C (180W) for 15 sec and second at 90 °C (45W) for 55 sec. Then, the resin was washed four times with DMF. Coupling reactions were carried out at two stages. First at 75 °C (155W) for 15 sec and second at 90 °C (30W) for 110 sec. The Fmoc-L-His(Trt)-OH was coupled at RT for 2 min and then at 50 °C for 8 min. To provide sufficient mixing during coupling and deprotection the vessel with resin and reagents was bubbled with nitrogen (repeating sequence - bubble on for 2 sec, off for 3 sec). The elongation of the peptide chain was carried out in consecutive cycles of deprotection and coupling. Peptides were cleaved from the resin using one of the mixtures (A) trifluoroacetic acid (TFA; Apollo Scientific, Denton, UK), triisopropylsilane (TIS, Iris Biotech GmbH, Marktredwitz, Germany), and deionized water (95:2.5:2.5, v/v/v), (B) TFA, TIS, phenol (Merck, Darmstadt, Germany), and deionized water (92.5:2.5:2.5:2.5, v/v/v/v) for 1.5 h with agitation. Mixture (B) was used with peptides containing tryptophan or tyrosine residue, and mixture (A) with the remaining peptides. Then, the compounds were precipitated with cooled diethyl ether (POCH, Avantor, Gliwice, Poland) and lyophilized. The compounds were purified by RP-HPLC. All purifications were carried out on a Waters X-Bridge BEH C18 OBD Prep column (19 × 100 mm, 5 μm particle size, 130 Åpore size). UV detection at 214 nm was used. Compounds were eluted with a linear 10-70% acetonitrile (ACN for HPLC-gradient grade; POCH, Avantor, Gliwice, Poland) gradient in deionized water over 60 min. The mobile phase flow rate was 10.0 mL/min. Both eluents contained 0.1% (v/v) of TFA. Pure fractions (> 95%, HPLC) were collected and lyophilized. The identity of all compounds was confirmed by mass spectrometry (ESI-MS; Waters Alliance e2695 system with Acquity QDa detector; Waters, Milford, MA, USA). All of the synthesized peptides have the C-terminal amide group.

**Organisms and antimicrobial assay.** Verification of antimicrobial activity of the synthesized analogs was conducted on reference strains of *E. coli* ATCC 25922 and ATCC 43827 in the first phase of the experimental validation, and *S. aureus* ATCC 33591, *E. coli* ATCC 25922, *E. coli* ATCC 43827, *A. baumannii* ATCC BAA 1605, *P. aeruginosa* ATCC 9027 in the second phase.

All the strains were stored at -80°C in Roti®-Store cryo vials (Carl Roth GmbH, Germany) and before the tests were transferred into fresh Mueller-Hinton Medium (BioMaxima, Poland) and incubated for 24 h at 37 °C. Fresh cultures were used in antimicrobial assay. The MIC values were evaluated by the broth microdilution method according to Clinical and Laboratory Standards Institute Protocol[86]. For this purpose, initial inoculums of bacteria ($0.5 \times 10^5$ colony forming unit (CFU)/mL) in Mueller-Hinton Broth were exposed to the ranging concentrations of compounds (1-512 μg/mL) and incubated for 18 h at 37 °C. The experiments were conducted on 96-well microtiter plates, with a final volume of 100 μL. Inoculums densities were adjusted spectrophotometrically (Multiskan™ GO Microplate Spectrophotometer, Thermo Scientific) at the wavelength of 600 nm. The MICs were taken as the lowest drug concentration at which a visible growth of microorganisms was inhibited[87]. Moreover MBC was evaluated. After MIC examination the samples from MIC well and two higher concentrations were seeded on the agar plates. Section with no visible growth of

bacterial colonies was reported as the MBC for each strain and compound. All experiments were conducted in triplicate. The positive (proper growth of bacteria) and negative (sterility) controls were included.

**Hemolysis assay.** The hemolysis assay was conducted according to the procedure described previously[88]. Sheep red blood cells (Defibrillated Sheep Blood, Graso, Poland) were used. Serial dilution of peptides (1-512 μg/mL) was conducted in PBS on 96-well plates. Then the solution of RBCs was added to each well (100 μL with a 4% concentration of erythrocytes (v/v). The control wells for 0 and 100% hemolysis were also prepared: RBCs suspended in PBS and 1% of Triton-X 100, respectively. After that the plates were incubated for 60 min at 37 °C and then centrifuged at 800 × g for 10 min at 4 °C (Sorvall ST 16R Centrifuge, Thermo Scientific). The supernatant was carefully resuspended to new microtiter plates and the release of hemoglobin was measured at 590 nm (Multiskan™ GO Microplate Spectrophotometer, Thermo Scientific). Hemolysis for each concentration was calculated as the percentage of control with 100% hemolysis. All experiments were conducted in triplicate. The final values of HC50 were computed using ic50.tk tool.

**Reporting summary**
Further information on research design is available in the Nature Portfolio Reporting Summary linked to this article.

## Data availability
The data used for training HydrAMP can be found at https://github.com/szczurek-lab/hydramp. The datasets used for evaluation of external classifiers can be found in the same repository and as Supplementary Data 8 and 9. Source data are provided with this paper. The datasets used in the manuscript can be downloaded from the Zenodo repository [https://zenodo.org/record/7420278#.ZBCn5ZHMKUk].

## Code availability
The HydrAMP source code, and the scripts for generation of the results can be found at https://github.com/szczurek-lab/hydramp. Free web-service is available with all the functionalities can be accessed at https://hydramp.mimuw.edu.pl/. The source code version used in the manuscript can be downloaded from the Zenodo repository [https://zenodo.org/record/7420189#.ZBCo4JHMKUk].

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

## Acknowledgements

We would like to thank NVIDIA for supporting this work and providing compute power used in MD simulations. This research was carried out with the support of the Interdisciplinary Centre for Mathematical and Computational Modelling University of Warsaw (ICM UW) under computational allocation no G88-1234.

## Author contributions

P.Sz., M.Mo., T.G. and E.S. conceived the project and methodology. P.Sz. curated the data. P.Sz., M.Mo. and T.G. implemented the model. P.Sz. performed the computational analysis. M.B., W.K., K.S. and D.N. performed wet lab experiments. M.Mi. and P.Se. performed computer simulations and resulting summaries of the novel peptides. R.J. performed benchmarking of the classifiers. J.S. and T.G. created the web service. P.Sz. and M.Mi. prepared the figures. E.S. supervised the study. P.Sz., M.Mo, and E.S. wrote the manuscript, with contributions and critical feedback from all authors.

## Competing interests

The authors P.Sz., M.Mo., T.G., R.J., M.B., D.N., M.Mi, P.Se, W.K, and E.S have issued a patent for the newly generated AMPs under the patent number P.443243. The other authors declare no competing interests.
