## [Peer Review File · Nature Communications]

HydrAMP: a deep generative model for antimicrobial peptide discoveryREVIEWER COMMENTS

Reviewer #1 (Remarks to the Author):

The manuscript details the development of HydrAMP as a deep generative model that addresses a critical challenge for antimicrobial peptide discovery. The capabilities of the newly generated model incorporating a conditional variational autoencoder that is simultaneously trained to follow the physicochemical properties of designed peptides while capturing their antimicrobial properties are articulated, explained, and showcased in detail. The competitive advantages of HydrAMP in peptide analog generation when compared with similar models is also defined and discussed. Such an approach can be valuable, reproducible and broadly applicable in multiple discovery interventions chemotypes and targets other than AMPs. Despite the conceptually strong elements, the robust computational design as well as implementation, the model validation is limited to synthetic chemistry, MIC experiments and in silico modeling to capture antimicrobial efficacy. The wet lab experiments are comprehensively described, and the reference *E. coli* strains are appropriate, but this part appears incomplete. Although it is only alluded to in the discussion the value of every tool for discovery would add value on AMPs peptides in tackling antimicrobial resistance. This at minimum requires biological validation beyond MIC values (Minimal Bactericidal Concentration, time kill assays, resistance phenotypic analysis with susceptible and resistant strains of additional species to identify their ability to target specific pathogens and/or dissect basic resistance mechanisms. There is clear notion in the discussion section as a future development, but a more extensive biological platform for validation will make this model more relevant, self-sustained and the manuscript acceptable for publication. For example, focusing on Enterobacteriaceae (*E. coli*, *Klebsiella*) and perform that more than minimal biological analysis. The manuscript is also clearly written but minor typos can be identified (eg ATCC instead of ATC strain Table 1, *Escherichia coli* to be introduced and then abbreviated to *E. coli*, italicized species name). The references should be accordingly enriched upon expansion of the biological piece and should consider similar state of the art interventions *Nat Biomed Eng* . 2021 Jun;5(6):613-623. doi: 10.1038/s41551-021-00689-x. Epub 2021 Mar 11

Reviewer #2 (Remarks to the Author):

The authors present a new method, HydrAMP, for antimicrobial peptide generation. The generative model of HydrAMP is a decoder, which produces a distribution over peptides in order to mimic the synthesis of the antimicrobial peptides in biological experiments. HydrAMP is the first model trained and used to identify active analogs of non-AMP prototypes. During training, they introduce an additional encoder to provide a variational approximation to help optimize the decoder. Their method is based on the conditional VAE model, which provides the foundation for an unconstrained generation. The method applied a two-fold regularization: the Jacobian disentanglement regularization to encourage disentanglement between the latent variable and a pair of conditions, and latent reconstruction regularization for better latent variable preservation properties.

This manuscript is nicely written. The idea of using a two-fold regularization to measure properties provided by a pair of conditions and imposes the relation of encoder and decoder in the model is interesting. And the proposed method, HydrAMP, has achieved great performance in a fraction of positives and negatives, the distribution of the probability of being AMP for negative prototypes and their analogs that met the discovery criteria, the distribution of probabilities of not being highly active for negative prototypes and their analogs that met the discovery criteria.

However, I have the following major concerns.

- (1) To improve clarity, the authors should explain why they choose exactly AMP and MIC classifiers.
- (2) Also, there is not much evidence that HydrAMP performs better in positive discovery than existing methods.
- (3) Moreover, the authors will need to compare with more autoencoders before they claim that HydrAMP has advantages over existing tools.
- (4) I am concerned about the lack of methodological novelty of the model, as cVAE based

antimicrobial peptide generation tools already exist, and HydrAMP does not perform much better than these methods. The author needs to describe HydrAMP's effective innovations more in the methodology section to demonstrate the breakthroughs made by the model.

(5) In figure 2 a, HydrAMP has close performance with the other two methods, and the discovery of AMP is considered as the key function of this generative model. I wonder if the authors can make more breakthroughs about it.

(6) In section 2.2, HydrAMP only compared with the other two models. However, there are still some methods based on LSTM or variational autoencoder. The authors are suggested to do more performance comparison experiments to enhance the persuasiveness of the paper.

(7) Authors mentioned they use three regularization terms, including Jacobian disentanglement regularization, and they all seem to be used correctly, but I wonder if there is any better alternative choice for regularization?

Reviewer #3 (Remarks to the Author):

Title: HydrAMP: a deep generative model for antimicrobial peptide discovery

The manuscript entitled "HydrAMP: a deep generative model for antimicrobial peptide discovery" proposes HydrAMP, a conditional variational autoencoder that intends to learn a lower-dimensional and continuous space of peptides' representations and captures their antimicrobial properties. HydrAMP was compared to other approaches in generating peptides, either de novo, or by analogue discovery. Manuscript unfortunately needs really improvement and some suggestions were here providing in order to improve the manuscript quality.

Suggestions

1. The authors present a new tool for rational AMP design. The article presents an interesting approach to generate new peptide sequences that possible are more potent and likely to have antimicrobial activity. However, in order to prove that, authors must test much more than a dozen of peptides. In fact, it is considering a real result if at least 200-300 peptides were tested. among all the analogs tested (four for Pexiganan and five for Temporin-A), six presented results that corroborate the predictions. Although the data seems promising, it would be interesting authors definitively need to synthesize more analogs to give robustness to their data. Authors could use the SPOT technology (please check Hilpert et al., 2007; 10.1038/nprot.2007.160.). Moreover, authors also need to tested the generated peptides against more strains of E. coli, including clinical isolates.
2. Negative bank used to feed the algorithm is always a real challenge for AMPs prediction. It seems that authors used several peptides that do not show an AMP flag activity, including the filters antimicrobial, antibiotic, antiviral, antifungal, effector, excreted. I understand that this was an important step, but unfortunately this is not sufficient in my opinion. Another option is to includes the presence of endogenous E. coli peptides that probably it will not affect E. coli cells.
3. Molecular dynamics data showed the potential of Hydraganan to penetrate the cell membrane. However, a longer simulation (μ s) would be interesting for the performed analyses. In addition, in silico results should be analyzed more carefully. Despite indicating the ability of the peptide to penetrate the membrane, in a biological system, several not simulated factors can influence the behavior of the molecule including salts, concentration, pH and the number of AMPs molecules. These concerns should be added to the text or taking in care at simulation.
4. In line 37 the period: "[...] aiming at increasing the resulting amphiphacity and/or charge" could be rephrased. Amphiphacity and charge are not the only targets of rational design.
5. In line 45 the term "active" should be replaced to avoid any misunderstanding once antimicrobial peptides are considered active already.
6. The link (L. 88) for the webserver is down.
7. The molecular models of the peptides should be presented somewhere. They are much more important than Figures 1 and 2, that could be easily added as supplementary material.
8. Authors are invited to add the training sequence datasets.
9. Authors must move the wet-lab experimental methods section for the main text.
10. Finally, and not lees important, authors are invited to test all peptides to mammalian cells that

could be erythrocytes or macrophages. It is important to differentiate between AMPs and peptides capable of destruct membranes with low selectivity. This will give more information about the peptide yielded by this novel in silico tool.

REVIEWER COMMENTS

We are very grateful for the insightful comments from all reviewers. We believe this revision has greatly improved our manuscript.

Reviewer #1 (Remarks to the Author):

The manuscript details the development of HydrAMP as a deep generative model that addresses a critical challenge for antimicrobial peptide discovery. The capabilities of the newly generated model incorporating a conditional variational autoencoder that is simultaneously trained to follow the physicochemical properties of designed peptides while capturing their antimicrobial properties are articulated, explained, and showcased in detailed. The competitive advantages of HydrAMP in peptide analog generation when compared with similar models is also defined and discussed. Such an approach can be valuable, reproducible and broadly applicable in multiple discovery interventions chemotypes and targets other than AMPs.

Despite the conceptually strong elements, the robust computational design as well as implementation, the model validation is limited to synthetic chemistry, MIC experiments and in silico modeling to capture antimicrobial efficacy. The wet lab experiments are comprehensively described, and the reference *E. coli* strains are appropriate, but this part appears incomplete. Although it is only eluded in the discussion the value of every tool for discovery would add value on AMPs peptides in tackling antimicrobial resistance. This at minimum requires biological validation beyond MIC values (Minimal Bactericidal Concentration), time kill assays, resistance phenotypic analysis with susceptible and resistant strains of additional species to identify their ability to target specific pathogens and/or dissect basic resistance mechanisms.

There is clear notion in the discussion section as a future development, but a more extensive biological platform for validation will make this model more relevant, self-sustained and the manuscript acceptable for publication. For example, focusing on Enterobacteriaceae (*E. coli*, *Klebsiella*) and perform that more than minimal biological analysis.

We are very grateful for this comment. We acknowledge the need for extensive biological validation. To improve the quality of our paper from the biological perspective, we extended the set of examined peptides and bacterial strains and we measured the activity of 24 new generated analogues and two previously identified active peptides, together with their prototypes (in total, 31 peptides) against:

- *Escherichia coli* ATCC 25922
- *Escherichia coli* ATCC 43827
- *Staphylococcus aureus* ATCC 33591
- *Acinetobacter baumannii* ATCC BAA 1605
- *Pseudomonas aeruginosa* ATCC 9027

Klebsiella pneumoniae was out of reach due to safety concerns. The analyzed *S. aureus* strain is methicillin-resistant, while *A. baumannii* ATCC BAA 1605 is resistant to multiple antibiotics. As such, in the revised version of the manuscript we analyze a broad group of

pathogens, including both susceptible and resistant strains, as well as gram-positive and gram-negative. Apart from MIC measurements, we now also measure Minimal Bactericidal Concentration (MBC) as well as toxicity, by measuring HC50 on sheep erythrocytes.

In order to increase the likelihood of the generated candidates being active against a wide spectrum of pathogens we devised an additional procedure of preselection of the candidates generated by HydrAMP, based on predictions of auxiliary classifiers and simulations. The new preselection procedure is described in **Section 2.7**. We have also added a new subsection where we describe the extended experimental validation (**Methods 5.2 and 5.3**), we describe its results in a **new Section 2.8** and summarize them in a **new Table 2** as well as **Supplementary Table S9**.

We believe that the extended microbiological validation has largely improved our manuscript.

The manuscript is also clearly written but minor typos can be identified (eg ATCC instead of ATC strain Table 1, *Escherichia coli* to be introduced and then abbreviated to *E. coli*, italicized species name).

The typos were fixed in the revised version. The *Escherichia coli* was introduced and then abbreviated to *E. coli*, with italicized species name. Please note that the ATCC is a valid name of an *E. coli* strain (ref. <https://www.atcc.org/products/25922>).

The references should be accordingly enriched upon expansion of the biological piece and should consider similar state of the art interventions Nat Biomed Eng . 2021 Jun;5(6):613-623. doi: 10.1038/s41551-021-00689-x. Epub 2021 Mar 11

Thank you for this comment. We have enriched the references and made a comprehensive review of 13 state-of-the-art methods for AMP generation and summarized them in a new **Supplementary Table S1** and in the **Introduction**. We carefully checked whether the authors made the code, trained model, training data, or a sample of generated peptides available. We also checked whether the results were at all validated experimentally, or computationally. In the case when the citation to any of these state-of-the-art methods was missing we added it to the manuscript.

In particular we also refer to Nat Biomed Eng . 2021 Jun;5(6):613-623. doi: 10.1038/s41551-021-00689-x. Epub 2021 Mar 11, although we note that this paper was already cited in the original version of the manuscript. Please note also that our success rate in experimental validation (54%) by far exceeded the one by this model (10%; see **Results Section 2.8**)

Reviewer #2 (Remarks to the Author):

The authors present a new method, HydrAMP, for antimicrobial peptide generation. The generative model of HydrAMP is a decoder, which produces a distribution over peptides in order to mimic the synthesis of the antimicrobial peptides in biological experiments. HydrAMP is the first model trained and used to identify active analogs of non-AMP prototypes. During training, they introduce an additional encoder to provide a variational approximation to help optimize the decoder. Their method is based on the conditional VAE model, which provides the foundation for an unconstrained generation. The method applied a two-fold regularization: the Jacobian disentanglement regularization to encourage disentanglement between the latent variable and a pair of conditions, and latent reconstruction regularization for better latent variable preservation properties.

This manuscript is nicely written. The idea of using a two-fold regularization to measure properties provided by a pair of conditions and imposes the relation of encoder and decoder in the model is interesting. And the proposed method, HydrAMP, has achieved great performance in a fraction of positives and negatives, the distribution of the probability of being AMP for negative prototypes and their analogs that met the discovery criteria, the distribution of probabilities of not being highly active for negative prototypes and their analogs that met the discovery criteria.

However, I have the following major concerns.

(1) To improve clarity, the authors should explain why they choose exactly AMP and MIC classifiers.

Thank you for this comment. The reason why we have decided to use two separate classifiers is that we make a separation between AMPs and active AMPs. In this work, we understand AMPs as all the peptides that are deposited as such in antimicrobial peptide databases such as DBAASP or CAMP. In turn, we use the term “active” referring to peptides which have MIC < 32 against *E. coli*. We focus on *E. coli* as it is the bacteria for which there were the most MIC values measured and deposited in databases. The AMP versus active AMP distinction is important, as there exist multiple AMPs, which are considered AMP and are not active against *E. coli*. For example, Stigmurin is active against *S. aureus* ATCC 29212 (MIC = 8.68 μ M), but inactive against *E. coli* ATCC 25922 (MIC >139 μ M; <https://dbaasp.org/peptide-card?id=8199>).

Taken together, this motivates the use of two classifiers, where the AMP classifier is trained on all the available AMP sequences, and MIC classifier is trained on a smaller subset of sequences with MIC measurements against *E. coli* strains. Incorporating the predictions of both these classifiers helps HydrAMP to generate peptides which are both AMP and active.

We have improved our explanation in **Section 4.2**. In addition, we have also benchmarked and explained the behavior of existing AMP classifiers in terms of distinguishing between active and inactive peptides. The benchmarking is reported in **Supplementary Text S2**, **Supplementary Table S5**, **Supplementary Figure S4** and mentioned in **Section 2.7**.

(2) Also, there is not much evidence that HydrAMP performs better in positive discovery than existing methods.

Thank you for this comment. First, we acknowledge that the use of the phrase “positive discovery” might have been confusing to the reader, possibly creating a false impression of what this task means and falsely underlining its importance. We realized that also other nomenclature referring to the tasks was unclear. To clarify this, in the revised version of the manuscript we use the following terms:

- *Baseline discovery/improvement discovery* when referring to criteria that generated analogues may satisfy. The baseline discovery criterion pertains to the generated peptide having desired properties with high probability: probability of being AMP $P_{M_AMP} > 0.8$ and probability of low MIC $P_{M_MIC} > 0.5$. In contrast, the improvement discovery criterion is satisfied when the generated peptide strictly improves upon the properties of the prototype peptide.
- *Analogue/unconstrained generation* when referring to the two generation types that HydrAMP offers (analogue meaning based on a prototype peptide, while unconstrained meaning *de novo*);
- *Positive/negative mode*, referring to generation of either positive (AMP and active) or negative (not AMP and inactive) peptides in the unconstrained generation;

Second, regarding the actual baseline discovery results. We assume that in this comment the reviewer is referring to Figure 2a. In the original manuscript, a mistake was made during the preparation of the figure, where AMP values were replaced with MIC (the mistake was not in the model itself, but rather in checking which criteria the generated peptides satisfied). This mistake is now corrected.

We also extended the figure to include results for HydrAMP with three different temperature parameter values. Additionally, in the revised version of this figure HydrAMP is compared with one more model, Joker, which is the only method other than HydrAMP or its ablated versions (Basic and PepCVAE) that is equipped with analogue generation functionality. In the corrected and extended version of **Figure 2a**, HydrAMP performs outstandingly well. Its stellar performance is demonstrated also in other panels of this Figure, which refer to improvement discovery. Additionally, there is clear evidence that HydrAMP performance is superior when compared to Joker. These results are now described in improved **Section 2.2**.

(3) Moreover, the authors will need to compare with more autoencoders before they claim that HydrAMP has advantages over existing tools.

We prepared a comprehensive summary of all available, previously published methods for AMP generation (see **Supplementary Table S1** and **Introduction**). We carefully checked whether the authors made the code, trained model, training data, or a sample of generated peptides available, making comparison to those methods possible. Based on this table, we chose and compared HydrAMP with four methods that perform unconstrained generation (AMP-LM, AMP-GAN, Dean-VAE, Muller-LSTM). Those four methods span a variety of frameworks, including variational autoencoder (Dean-VAE), generative adversarial network (AMP-GAN), linguistic model (AMP-LM), and LSTM (Muller-LSTM). The unconstrained

generation results are described in **Section 2.3** in the revised version of the manuscript and on **Figure 3** and **Supplementary Figure S1**.

In terms of analogue generation, we compared ourselves with Joker, the only other method that can take an existing input as a template for generation (apart from previously evaluated Basic and PepCVAE). This comparison is described in **Section 2.2** and visualized in **Figure 2** and **Figure 4**.

(4) I am concerned about the lack of methodological novelty of the model, as cVAE based antimicrobial peptide generation tools already exist, and HydrAMP does not perform much better than these methods. The author needs to describe HydrAMP's effective innovations more in the methodology section to demonstrate the breakthroughs made by the model.

The **revised Figures 2, 3, 4**, and **Supplementary Figure S1** convincingly show that HydrAMP clearly outperforms a wide range of previous methods (together, seven). In terms of theoretical advantages, HydrAMP is the only approach that is explicitly trained for positive and negative modes in analogue generation (**Figure 1b**), it utilizes newly introduced latent Jacobian regularization, and leverages the modeled Decoder distribution for generation. We now extended the paragraph summarizing HydrAMP novelties in the **Discussion** section. It is also more evident after adding explicit descriptions of how generation in Basic and PepCVAE models is implemented (**Supplementary Text S1**).

(5) In figure 2 a, HydrAMP has close performance with the other two methods, and the discovery of AMP is considered as the key function of this generative model. I wonder if the authors can make more breakthroughs about it.

We have extended the comparison of our model to other methods in **Section 2.2** and **2.3**, as well as corrected and revised the **Figure 2** and **Figure 3** as we described in the response to points (2), (3), and (4). We emphasize that in this comparative evaluation, HydrAMP clearly outperformed other methods in all tasks.

(6) In section 2.2, HydrAMP only compared with the other two models. However, there are still some methods based on LSTM or variational autoencoder. The authors are suggested to do more performance comparison experiments to enhance the persuasiveness of the paper.

This comment encouraged us to perform more extensive comparison to other available methods, which we collected in **Supplementary Table S1** and described in the Introduction. Please note that the reproducibility of existing methods is significantly limited as only 9 out of 13 methods share the code for their method, and only 4 of those provide the trained model. Given those difficulties, we managed to run only one of those methods (AMP-GAN), and collected generated peptides for 4 other methods (AMP-LM, Joker, DeanVAE, Muller-RNN). In this set we have a method based on VAE (Dean-VAE), a GAN (AMP-GAN), a genetic algorithm (Joker), and LSTM-based method (Muller-RNN). We repeated the experiments and we reported them in the revised version of the paper on **Figure 2**, **Figure 3**, **Figure 4**, as well as **Supplementary Figure S1**. The description of the results can be found in **Section 2.2** and **Section 2.3**.

(7) Authors mentioned they use three regularization terms, including Jacobian disentanglement regularization, and they all seem to be used correctly, but I wonder if there is any better alternative choice for regularization?

This is a really wide and open question, as there are numerous possible ways in which deep neural network models could be regularized. Please note that apart from the novel Jacobian disentanglement regularization, we also use a more conventional approach i.e. drop-out. We also treat the negative mode in unconstrained generation as a form of regularization as now described in the **Discussion**.

One could potentially use one of the generally applicable and recently introduced methods of regularization of deep learning methods. For example, we could apply weight regularizers and batch norm for the neural network architecture. Other regularization methods might be applied to further impose the Encoder distribution to be similar to the prior distribution, e.g. MMD regularization or adversarial regularization. Finally, one could apply the adversarial regularization to the generated peptides by training an additional model that similarly to GAN discriminators is trained to discriminate between true peptides and generated peptides.

However, in this work we instead deliberately focus on solutions tailored for the new model for AMP generation. The newly introduced Jacobian disentanglement regularization is devised specifically for this task and this type of framework, i.e. cVAE. The comparison with PepCVAE and Basic, which lack Jacobian disentanglement regularization, can be treated as an ablation study. In a number of tasks, HydrAMP outperforms PepCVAE and Basic both in terms of the P_{M_AMP} and P_{M_MIC} distributions (**Figure 2c,d**), as well as the fraction of prototypes for which the models managed to generate analogues that meet a specific criterion (**Figure 2a,b**). We focused on assessing the generative performance of the model instead of comparing various regularizers. Such benchmarking of regularizers would make a separate paper on its own.

Reviewer #3 (Remarks to the Author):

Title: HydrAMP: a deep generative model for antimicrobial peptide discovery

The manuscript entitled “HydrAMP: a deep generative model for antimicrobial peptide discovery “ proposes HydrAMP, a conditional variational autoencoder that intends to learn a lower-dimensional and continuous space of peptides’ representations and captures their antimicrobial properties. HydrAMP was compared to other approaches in generating peptides, either de novo, or by analogue discovery. Manuscript unfortunately needs really improvement and some suggestions were here providing in order to improve the manuscript quality.

Suggestions

1. The authors present a new tool for rational AMP design. The article presents an interesting approach to generate new peptide sequences that possible are more potent and likely to have antimicrobial activity. However, in order to prove that, authors must test much more than a dozen of peptides. In fact, it is considering a real result if at least 200-300 peptides were tested. among all the analogs tested (four for Pexiganan and five for Temporin-A), six presented results that corroborate the predictions. Although the data seems promising, it would be interesting authors definitively need to synthesize more analogs to give robustness to their data. Authors could use the SPOT technology (please check Hilpert et al., 2007; 10.1038/nprot.2007.160.). Moreover, authors also need to tested the generated peptides against more strains of E. coli, including clinical isolates.

Thank you for this comment. To improve the quality of our paper from the biological perspective, in the revised version, we largely increased the number of synthesized and tested peptides, as well as the scope of the tests. In this experimental round, we tested 24 new generated analogues and two previously identified active peptides, together with their prototypes (in total, 31 peptides) on an extended set of examined strains:

- *Escherichia coli* ATCC 25922
- *Escherichia coli* ATCC 43827
- *Staphylococcus aureus* ATCC 33591
- *Acinetobacter baumannii* ATCC BAA 1605
- *Pseudomonas aeruginosa* ATCC 9027

The analyzed *S. aureus* strain is methicillin-resistant, while *Acinetobacter baumannii* ATCC BAA 1605 is resistant to multiple antibiotics. As such, in the revised version of the manuscript we analyze a broad group of pathogens, including both susceptible and resistant strains, as well as gram-positive and gram-negative bacteria. Apart from MIC measurements, we now also measure Minimal Bactericidal Concentration (MBC) as well as toxicity against sheep erythrocytes (HC50).

To pick the best candidates for this extended experimental validation, we applied an additional procedure based on filtering and ranking of the candidates generated by HydrAMP using external classifiers and molecular dynamics simulations.

The success rate of HydrAMP in experimental validation is exquisite. Our main competition, CLaSS by Das et al. 2021 (<https://doi.org/10.1038/s41551-021-00689-x>) tested 20 candidate peptides, only two of them showed low MIC values (MIC < 32 µg/mL). In comparison, out of 24 generated by HydrAMP and synthesized peptides, 16 were proven active against *E. coli* ATCC 25922.

We have added a new subsection where we describe the extended experimental validation, we describe its results in a **new Section 2.8** and summarize them in **new Table 2**. The preselection procedure with auxiliary classifiers and MD simulations is described in newly added **Section 2.7**, and the evaluation of these classifiers is shown in **Supplementary Table S5**.

Regarding the number of tested peptides, please note that the main subject of this work is the novel deep generative model. While we put effort into validating our results via wet-lab experiments, this is not a standard in the community working on computational approaches for AMP design. We prepared a table (**Supplementary Table S1**) which summarizes existing approaches and their scope of both experimental and computational validation. Out of 13 evaluated methods, 5 of them did not perform any kind of wet lab validation. Among the ones which measured at least MIC, the number of tested peptides varied between 10 to 30 peptides. There was one notable exception, namely the work of Yoshida et al. 2018 (<https://doi.org/10.1016/j.chempr.2018.01.005>). Here, the authors synthesized as many as 141 peptides. Taken together, the total number of tested peptides in our work (33 analogues and 6 prototypes in total in the two experimental rounds) is in the very high range, as compared to previous studies.

We would like to thank you for suggesting the SPOT technology. However, after careful consideration, we have decided not to perform this type of experiment. In the discussed summary of methods only one model uses SPOT technology for experimental validation (Porto et al. 2018, <https://doi.org/10.1016/j.bbagen.2018.06.011>). The SPOT technology cannot guarantee sufficiently pure peptides for antimicrobial testing. Indeed, some authors reported significant impurities (Kramer et al. 1999, <https://doi.org/10.1034/j.1399-3011.1999.00108.x>). Purification of such a mixture can be complicated due to a small amount of peptide (µmol) and e.g. concomitant truncated peptides. Furthermore, side products can influence bacterial growth and therefore interfere with the tested peptide. Evaluation of antimicrobial activity of peptides synthesized by proposed SPOT technology bases on some assumptions and simplifications that can undermine accuracy of results. All in all, this approach may give preliminary results of antimicrobial activity (Hilpert K; Hancock R. E. W 2007, <https://doi.org/10.1038/nprot.2007.203>), and finally need to be verified using peptides of known concentration and purity (synthesized on resin). Having this in mind, we decided to rely on a high quality, reproducible and clean method of peptide synthesis and validation, as described in **Methods Section 5**.

2. Negative bank used to feed the algorithm is always a real challenge for AMPs prediction. It seems that authors used several peptides that do not show an AMP flag activity, including the filters antimicrobial, antibiotic, antiviral, antifungal, effector, excreted. I understand that this was an important step, but unfortunately this is not sufficient in my opinion. Another

option is to include the presence of endogenous *E. coli* peptides that probably it will not affect *E. coli* cells.

We find this comment extremely valid for approaching AMP design with ML methods in general. We agree that constructing the negative dataset is a tricky task. In case of HydrAMP, we tried to circumvent the lack of data of experimental quality in a twofold manner:

- 1) We used an AMP classifier trained on both positive and negative training data. In this case, the positive dataset consists of all of the peptides deposited in various AMP databases. The negative dataset, as noted by the reviewer, consists of UniProt sequences of length not exceeding 25 amino acids excluding sequences which meet a number of criteria such as antibiotic, antiviral etc. In particular, sequences satisfying these criteria and exceeding the length 25 were cut to 25 amino acids.
- 2) We used a MIC classifier which predicts whether a peptide is active or not. Here, we used only sequences with reported MIC values, separating active AMP (MIC < 32) from inactive (MIC > 32).

HydrAMP uses both these classifiers to guide the generation and they are both weighted equally. While the former AMP classifier serves as a general filter based on a larger number of sequences, the latter MIC classifier provides a more granular distinction of positive and negative peptides.

We agree that it would be most favorable to use only experimentally verified sequences, most preferably including endogenous *E. coli* peptides as well. However, this approach is simply not feasible using deep learning methods, as the number of training samples would be too small. Notably, even without the experimentally verified negative set, HydrAMP still manages to generate both positive and negative analogues as demonstrated in Table 1. We now added this important point of negative set selection to the **Discussion**.

3. Molecular dynamics data showed the potential of Hydraganan to penetrate the cell membrane. However, a longer simulation (μ s) would be interesting for the performed analyses. In addition, *in silico* results should be analyzed more carefully. Despite indicating the ability of the peptide to penetrate the membrane, in a biological system, several not simulated factors can influence the behavior of the molecule including salts, concentration, pH and the number of AMPs molecules. These concerns should be added to the text or taken in care at simulation.

We acknowledge that protein-membrane systems exhibit slow relaxation, which makes it challenging to achieve satisfactory sampling of configuration space. At the same time, our tests indicate that the *S* parameter, being relatively coarse grained, tends to equilibrate within a few hundred nanoseconds (**Figures 5e, 5f** and **Supplementary Fig. S5**). Still, to provide for a safe margin, we extended all major MD production runs to 1 μ s.

We are also aware of the limitations of our simulation model in terms of its inability to represent rich variability of biological conditions. We note though, that the inclusion of a number of controllable factors would lead to combinatorial explosion of possible setups, making the entire simulation part unfeasible. Accordingly, in order to be able to effectively

screen 180 peptides, which was part of the newly added preselection procedure (**Section 2.7**), we decided to maintain a generic system representation. We validated the relevance of predictions using *S* based on a broader set of experimentally verified active and inactive peptides (9 cases, in addition to 2 previously considered). At the same time, we added a brief discussion of model limitations (**Section 2.6**) to explicitly make the readers aware of them.

4. In line 37 the period: “[...] aiming at increasing the resulting amphiphacity and/or charge” could be rephrased. Amphiphacity and charge are not the only targets of rational design.

Thank you for this comment. We have rephrased this sequence to: “usually aiming at increasing the resulting amphiphacity and/or charge”. We are aware that in the rational design process one could optimize other targets, such as the hydrophobic moment or the alpha-helix propensity, or decreasing the degradability.

5. In line 45 the term “active” should be replaced to avoid any misunderstanding once antimicrobial peptides are considered active already.

In this work we make a distinction between a peptide being an AMP and a peptide being active (having MIC < 32 against *E.coli* strain), using this terminology throughout the text as well as training two separate classifiers (one for being AMP and one for being active). To clarify this early in the text we elaborated on this subject in **Methods Section 4.2**. Notably, while all of the peptides deposited in databases such as DBAASP or CAMP are treated as AMP, only a subset of them show low MIC against *E. coli* and are thus considered active. The reason for introducing this division in both the notation and the experimental setup is that we are interested not only in AMPs, but specifically in peptides with low MIC values. To encourage our model to generate such peptides, we train the model to generate and recognize both active (MIC<32) and inactive (MIC>32) peptides.

6. The link (L. 88) for the webserver is down.

Thank you for spotting this. We have set up a redirection from http to https. Currently both links are active and the website is fully functional:

<https://hydramp.mimuw.edu.pl/>

<http://hydramp.mimuw.edu.pl/>

7. The molecular models of the peptides should be presented somewhere. They are much more important than Figures 1 and 2, that could be easily added as supplementary material.

We have added the molecular models as **Supplementary Figures S6-9**. We also provide molecular structures in an on-line repository

(https://drive.google.com/drive/folders/1DZrLo3iJibUrvQymeSxW_CkAxcsvQjTp; Molecular Structures)

Additionally, we describe three representative molecular structures that we found across all 180 peptides (information on all the peptides is collected in **Supplementary Table S8**) that we simulated and we compare them to structures that we were able to obtain from AlphaFold predictions (**Figure 6**).

8. Authors are invited to add the training sequence datasets.

The training data were available for download from the GitHub repository in the first version of the paper. Because of the size of the datasets, we made them available through DVC or Google Drive. For details, please refer to the README file in our repository: <https://github.com/szczurek-lab/hydramp>.

9. Authors must move the wet-lab experimental methods section for the main text.

We have moved the wet-lab experimental methods section from the Supplementary Material to **Section 5 in the Methods** in the main text.

10. Finally, and not less important, authors are invited to test all peptides to mammalian cells that could be erythrocytes or macrophages. It is important to differentiate between AMPs and peptides capable of destruct membranes with low selectivity. This will give more information about the peptide yielded by this novel in silico tool.

Thank you for this comment. Although the HydrAMP model does not take into account the toxicity, we decided to extend the experimental validation and we measured HC50 on sheep erythrocytes for 24 new analogues, 2 previously described analogues of Pexiganan, and 5 prototype peptides (31 in total). The results presented in **Table 2** show very low toxicity of the peptides generated by HydrAMP.

REVIEWER COMMENTS

Reviewer #1 (Remarks to the Author):

The manuscript has been improved dramatically including comprehensive articulation for rationalizing/optimizing the AMP selection model as well as expanded in "wet lab" experimentation as requested to support predictive modeling. Overall, it reads better, highlights the primary results for the validity of the model. Nevertheless, the structure and rationalization of the biological experiments, although are a valuable addition, lack a translational flowchart and a decision-making process e.g. species, strain, model selection. For example, it is puzzling that KPC raises biological concerns (is a conventional BSL-2 used pathogen) and 2 E. coli strains are selected, or hemolytic assay for blood erythrocytes is chosen as a primary toxicity assay. Typos and phrasing require further attention in definitions (what is the contribution of "against a specific strain of bacteria" has to offer in the MIC definition for an antimicrobial peptide? 2 E.coli strains are used, besides that there are bacterial species. Cut off selections for MIC (32 OR 100ug/ml respectively appear arbitrary). Discussion and data analysis eg Table 2 are there other patterns observed between Gram+ or Gram- species? Although only an ATCC MRSA strain was employed, it adds value to comment in susceptibility profile/pathogen also rather than peptide or structure.

Reviewer #2 (Remarks to the Author):

Thanks to the authors to addressing some of my concerns. While I really like the detailed explanation and comprehensive experiments, some settings in experiments might not be obvious for me. There are several points that I would deem important to address.

1. In section 2.1, the authors mentioned '25 amino-acid long, non-redundant sequences collected from UniProt', which means that the length of training data from UniProt is below 25. But according to statistics, nearly half of the existing AMPs have the lengths ranging from 25 to 50. What's the intuition for choose 25 as the threshold value, and it is necessary to declare to what extent this data improves the performance of the model, because there is not much explanation about this part of data.
2. The authors added 'HydrAMP benefits from an additional temperature parameter τ that controls the creativity of analogue generation.' However, it is unclear how this parameter works in the model. Further explanation/reference is needed.
3. The authors defined two distinct evaluation criteria: 'baseline discovery' and 'improvement discovery', which are very intuitive in this task. However, in addition to the comparison of the fraction of accepted analogues, how are the confidence scores of these accepted analogues? Are the sequences generated by HydrAMP have the highest average score among all the methods?
4. 'The resulting set of representative sequences contains only sequences that are longer than 25 amino acids' in section 4.1, maybe the authors want to say not longer?
5. Figure 1.b and c seem too simple, especially for the peptide sequence and the vector part. It is not easily to get the point of the figure at the first glance, for example: how does the loss backpropagate? The figure needs to be more apparent and vivid.
6. In 'Identification of antimicrobial peptides from the human gut microbiome using deep learning' by Ma et al. 2022, they chemically synthesized 216 sequences screened by their method, and 181 showing antimicrobial activity (a positive rate of >83%). Their method looks better in terms of quality and quantity, therefore, what's the advantages of HydrAMP over theirs?

Reviewer #3 (Remarks to the Author):

Manuscript it was improved and was acceptable in the present form

Revision 2

We are grateful to Reviewers' comments. We have improved the manuscript accordingly. Below we provide detailed answers to the comments in blue font.

Reviewer #1 (Remarks to the Author):

The manuscript has been improved dramatically including comprehensive articulation for rationalizing/optimizing the AMP selection model as well as expanded in "wet lab" experimentation as requested to support predictive modeling. Overall, it reads better, highlights the primary results for the validity of the model.

Nevertheless, the structure and rationalization of the biological experiments, although are a valuable addition, lack a translational flowchart and a decision-making process e.g. species, strain, model selection.

For example, it is puzzling that KPC raises biological concerns (is a conventional BSL-2 used pathogen) and 2 *E. coli* strains are selected, or hemolytic assay for blood erythrocytes is chosen as a primary toxicity assay.

Thank you for this comment! The experimental setup served first validation purposes and second was dictated by research questions regarding the comparative activity of the generated peptides against a range of different species/strains of bacteria. The motivation behind the selection of species and strains is now explained in the revised manuscript **Section 2.8**.

Regarding the experimental setup, we selected two *E. coli* strains to validate the hypothesis that HydrAMP can generate peptides that are active against the same species of bacteria as in its training data. Specifically, HydrAMP was trained on peptides with their MIC values measured exclusively against *E. coli* strains. The two selected strains of *E. coli* are widely used to determine the antimicrobial activity of novel compounds, including peptides, as reported in numerous previous studies (e.g., <https://pubs.rsc.org/en/content/articlepdf/2019/ra/c9ra00708c>; <https://academic.oup.com/jac/article/61/2/341/767006>; <https://www.ncbi.nlm.nih.gov/pmc/articles/PMC5167725/>; <https://www.ncbi.nlm.nih.gov/pmc/articles/PMC9410094/>; <https://www.nature.com/articles/s42003-022-03899-4>).

Moreover, *E. coli* ATCC 25922 is a CLSI (Clinical Laboratory Standards Institute) control strain for antimicrobial susceptibility testing. Here, the expected validation result was to obtain similar MIC results for both *E. coli* strains (+/- one dilution in the dilution series, e.g. MIC1 = 16 µg/mL, expected MIC2 = <8-32> µg/mL). This similarity was observed for most of the peptides, as reported in Table 2. This validation result is now discussed in the revised manuscript, **Section 2.8**.

Moreover, to investigate the difference between the activities of our peptides against *E. coli* to other gram negative bacteria, we included additional two gram negative strains: *Acinetobacter baumannii* ATCC BAA 1605 and *Pseudomonas aeruginosa* ATCC 9027. The summary of this comparison is now reported in the revised manuscript, **Section 2.8**.

Finally, to complete the overview of the activity of the novel peptides, we included a strain of gram positive bacteria - *S. aureus* ATCC 33591 - in the study. This allowed for an initial comparison of the activity of compounds between strains of gram negative and gram positive bacteria. A discussion of the observed activity profiles of our peptides on the gram positive versus gram negative bacterial strains is now added in **Section 2.8**.

To sum up, the baseline validation group of bacteria for our model are the strains of *E. coli*. The remaining strains serve as reference strains for comparative analysis - both between different species of gram-negative and gram-positive bacteria.

Given the fact that this selection of bacterial strains was enough for the planned validation and comparative analysis, we did not include any additional strains. In particular, we did not include *Klebsiella pneumoniae*. Indeed, we admit that *K. pneumoniae* is in the same biosafety group (BSL 2) as *S. aureus* ATCC 33591, *A. baumannii* ATCC BAA 1605, and *P. aeruginosa* ATCC 9027. However, adding it would contribute a fifth representative of the gram negative bacteria, thus, we did not expect it would bring much additional value to the already comprehensive experimental screen.

Such decisions are always difficult to make due to the fact that there are many interesting strains of clinical relevance. In our opinion, the pool of strains used is sufficient to achieve the set research goals.

The hemolysis assay for blood erythrocytes was chosen as a primary toxicity assay as it is standard practice for toxicity testing, as used in multiple previous studies (e.g., <https://www.ncbi.nlm.nih.gov/pmc/articles/PMC2592880/>, <https://link.springer.com/article/10.1134/S106816202103002X>, <https://www.nature.com/articles/ja20144#Sec2>, <https://www.pnas.org/doi/full/10.1073/pnas.1918427117#sec-1>, <https://www.sciencedirect.com/science/article/pii/S0300908420301322#appsec1>, <https://pubs.acs.org/doi/10.1021/acsomega.9b02278>). Indeed, erythrocytes are a good test model for the initial toxicity assessment. The cell membrane is the molecular target of cationic antimicrobial peptides. Bacterial and erythroid cells differ significantly in the structure of the cell membrane. Hemolytic activity is associated with damage to the erythrocyte membrane, which results in the release of hemoglobin from the cell. The hemoglobin extracted from the cell can be easily and quickly identified (spectrophotometer), and the absorbance can be related to the degree of membrane disruption. The advantage of this experiment is the ease of obtaining erythrocytes (mammalian blood) and the short duration of the study. In addition, one of the possible routes of administration of antimicrobial drugs is intravenous administration, where drug molecules come into contact with erythrocytes (e.g. lipopeptide - Daptomycin, Cubicin). Therefore, erythrocytes can be considered for their use in toxicity studies.

An alternative option would be to test the cytotoxicity of the peptides against cell lines (IC50). Some authors indicate that erythrocytes are more sensitive to the action of cationic antimicrobial peptides than cell lines (i.e. frequently used - HaCaT, HeLa) (<https://www.nature.com/articles/s41598-020-69995-9#Sec2>). Therefore, to some extent we can expect that if a peptide is not toxic to erythrocytes, it will also not be toxic to cell lines. Thus, our choice of experiment using red blood cells seems to be justified. This is now mentioned in **Section 2.8**.

Typos and phrasing require further attention in definitions (what is the contribution of "against a specific strain of bacteria" has to offer in the MIC definition for an antimicrobial peptide? 2 *E.coli* strains are used, besides that there are bacterial species).

Thank you for this comment. We have again re-read the paper looking for typos.

Regarding the phrase "against a specific strain of bacteria" in MIC definition. Since MIC is measured against specific bacterial strains, it can be different against different bacteria and different strains for the same peptide. Therefore, to maximize the size of the training set, we needed to pick the bacteria with the most abundant MIC measurements (and this happened to be *E. coli*). We have also averaged MIC measurements across *E. coli* strains. This is now clarified in the manuscript, **Section 2.1**. For the exact details see our data processing pipeline, **Section 4: Peptides with experimentally proven antimicrobial activity with measured MIC values**, https://github.com/szczurek-lab/hydramp/blob/master/scripts/dataset_preparation.ipynb.

Cut off selections for MIC (32 OR 100µg / mL respectively appear arbitrary).

Thank you for pointing this out. Indeed, it is difficult to choose a specific cut-off. Careful re-inspection of previous studies revealed that those studies considered a much more permissive threshold of 128 µg/mL, rather than the conservative threshold of 32 µg/mL that we assumed in the previous version of our manuscript. In particular, we determined that the threshold of 100 µg / mL was applied by Porto et al. only for the SPOT assay. However, for the microbiological assay measuring MIC they considered MIC = 128 µg/mL to be indicative of peptide's activity.

Once we reevaluated our results using the more common threshold of 128 µg/mL our method achieved as high an AMP success rate as 96%, exceeding the success rates reported by previous studies. This is now reported in **Section 2.8**.

Nevertheless, we agree with the Reviewer that all these cut-offs seem arbitrary. In particular, smaller MIC thresholds are more clinically relevant thresholds than 128 µg/mL. Therefore, we performed a comprehensive comparison of AMP success rates for smaller thresholds (**new Figure 7**). Again, we show that overall our method outperforms others in terms of AMP success rate, especially for the most conservative and challenging thresholds MIC = 8 and 16 µg/mL. This result is now reported in **Section 2.8**. We hope that in the future, other authors will adapt this way of reporting success rates, instead of sticking to a single cut-off.

Discussion and data analysis eg Table 2 are there other patterns observed between Gram+ or Gram- species? Although only an ATCC MRSA strain was employed, it adds value to comment in susceptibility profile/pathogen also rather than peptide or structure.

Indeed, this experimental design allows us to compare the activity of the newly generated peptides between the gram positive and gram negative species.

Inspection of the respective activity profiles indicated that the relative activity of a peptide against *S. aureus* versus the remaining bacterial strains varied from case to case and there was no clear tendency. In particular, gram positive bacteria were not always more sensitive to AMPs than gram negative bacteria, nor vice versa. For example, the new AMP called Rudyxin, was inactive against *S. aureus* (MIC=512 µg/mL), and showed activity against *E. coli* (ATCC 25922 MIC = 128 µg/mL, ATCC 43827 MIC = 4 µg/mL). In contrast, the AMP called Hydraganan-2, showed high activity against *S. aureus* (ATCC 33491 MIC=32 µg/mL), but was less active against *E. coli* (ATCC 29522 MIC = 128 µg / mL). There were also peptides highly active against both *S. aureus* and *E. coli*, such as Sophieganan (*E. coli* ATCC 25922 MIC = 4 µg/mL, *S. aureus* ATCC 33591 MIC = 4 µg/mL). This comparative discussion of the results is now included in the revised manuscript, **Section 2.8**.

Similar results can be found reported in the literature. For example, Fritsche T.R. et al showed that Omiganan shows higher activity against *S. aureus* strains (MIC <= 32 µg/mL) than against *P. aeruginosa* (MIC <= 256 µg/mL) (<https://academic.oup.com/jac/article/61/5/1092/848354>). On the other hand, Pexiganan shows higher activity against *A. baumannii* strains (MIC 2-4 µg/mL for 87 out of 113 strains) than against *S. aureus* (MIC 8-16 µg/mL for 425 out of 512 strains) (<https://www.ncbi.nlm.nih.gov/pmc/articles/PMC89207/>).

Reviewer #2 (Remarks to the Author):

Thanks to the authors to addressing some of my concerns. While I really like the detailed explanation and comprehensive experiments, some settings in experiments might not be obvious for me. There are several points that I would deem important to address.

1. In section 2.1, the authors mentioned '25 amino-acid long, non-redundant sequences collected from UniProt', which means that the length of training data from UniProt is below 25. But according to statistics, nearly half of the existing AMPs have the lengths ranging from 25 to 50. What's the intuition for choose 25 as the threshold value, and it is necessary to declare to what extent this data improves the performance of the model, because there is not much explanation about this part of data.

Thank you for this comment. The threshold of 25 amino-acids for the length of peptide sequences is due to two reasons: availability of training data and the ability of synthesizing such peptides in the lab.

As we show in a **new Supplementary Fig S10**, AMPs up to 25 amino-acids long constitute the majority of peptides with length up to 100 amino-acids 11131/19883; 56%. This amounts to 11131/20313 (55%) of all available AMP sequences. See also distribution of sequence lengths for all AMP peptides in the DBAASP database (<https://dbaasp.org/statistics/general>).

At the same time, our experience in the experimental lab tells us that a threshold of 25 amino acids is also an appropriate synthesizability cut-off. Our syntheses usually give satisfactory results (good purity of raw products, i.e. >60%) if they do not exceed 25 amino acid residues. Of course, the final synthesis result is very dependent on the particular amino acid sequence. Undoubtedly, the chances of obtaining the right products of satisfactory purity are greater in the case of shorter peptides than in the case of longer ones. This is evident and is due to the decreasing overall synthesis yield with increasing length of the peptide chain. The choice of this length threshold is now motivated in the revised manuscript, **section 4.1**.

2. The authors added 'HydrAMP benefits from an additional temperature parameter τ that controls the creativity of analogue generation.' However, it is unclear how this parameter works in the model. Further explanation/reference is needed.

We have added an additional explanation of how this parameter works in the model in **Section 2.1 and 4.6.2** of the main text. Additionally, the influence of this parameter is illustrated in Figure 2cd. Thank you for raising this issue.

3. The authors defined two distinct evaluation criteria: 'baseline discovery' and 'improvement discovery', which are very intuitive in this task. However, in addition to the comparison of the fraction of accepted analogues, how are the confidence scores of these accepted analogues? Are the sequences generated by HydrAMP have the highest average score among all the methods?

Thank you for this comment. We have now added a **new Figure 3**, illustrating the distributions of the probability of being AMP (P_M_AMP) and of being active (P_M_MIC) for the analogues that satisfy either the baseline or the improvement discovery criteria, and their either negative or positive prototypes, covering all four possible combinations. These comparisons again indicate that our model can improve peptides that are more difficult to improve (such that have initially lower P_M_AMP and P_M_MIC), than other models, and generates analogues with higher resulting P_M_AMP and P_M_MIC. This **new Figure** is now discussed in detail in **Section 2.2**

4. 'The resulting set of representative sequences contains only sequences that are longer than 25 amino acids' in section 4.1, maybe the authors want to say not longer?

Thank you for pointing this out. This typo was now fixed.

5. Figure 1.b and c seem too simple, especially for the peptide sequence and the vector part. It is not easily to get the point of the figure at the first glance, for example: how does the loss backpropagate? The figure needs to be more apparent and vivid.

Thank you for this comment. We **improved Figure 1** accordingly. First, in order to make the figure more vivid we have changed the color palette. Second, we have adjusted the representation of the peptide and vectors. In the revised version, a peptide is represented with an amino acid sequence as opposed to the visualization of the vector in the latent space.

Regarding backpropagation, we acknowledge that it is not straightforward to grasp how the loss backpropagates based on Figure 1 alone. The details of how the loss function is optimized are described in detail in **Supplementary Methods Section S4** and **Methods Section 4.3** and are difficult to visualize. Instead, this figure illustrates the terms in the loss function. To better explain the visual notation used in the Figure, we expanded its caption with additional details.

6. In 'Identification of antimicrobial peptides from the human gut microbiome using deep learning' by Ma et al. 2022, they chemically synthesized 216 sequences screened by their method, and 181 showing antimicrobial activity (a positive rate of >83%). Their method looks better in terms of quality and quantity, therefore, what's the advantages of HydrAMP over theirs?

Thank you for pointing us to the paper of Ma et al, 2022. This model is now cited in the Introduction of the revised manuscript, together with other QSAR models.

However, we note that the positive rate depends on the used threshold. Moreover, the authors of this paper measured the activity of peptides in different units (μM) than in our work ($\mu\text{g/mL}$). In their experiments, they considered a peptide active based on a threshold of 60 μM .

To be able to compare our results to the results of this model in terms of the reported success rate, we recalculated our MIC, measured in $\mu\text{g/mL}$ to μM (**Supplementary Table**

S11). The recalculations took into account counterion content and are based on our previous research, suggesting that it is possible to make an estimate of the peptide and its counterion content in the sample (<https://pubmed.ncbi.nlm.nih.gov/29307075/>). The assumption is that for each peptide molecule there are as many counterion molecules (i.e. TFA-) as there are basic moieties present. It should be taken into account that this is a significant simplification. The exact conversion should be based on the measurement of the peptide and counter- ion content in the sample.

Using the recalculated values and the same threshold of 60 μM as Ma et al., we obtained a higher AMP success rate (96%) for our model than the one reported by the authors (84%). This comparison is now reported in the revised manuscript, **Section 2.8**.

Moreover, importantly, the model of Ma et al. is not a generative model. It only screens a given database of input peptides and selects the most promising ones for validation. Thus, it is inherently unable to produce new peptides that did not exist in its input database.

In summary, HydrAMP outperforms Ma et al. in AMP success rate and offers a much more sophisticated methodology for generation of novel peptides.

Reviewer #3 (Remarks to the Author):

Manuscript it was improved and was acceptable in the present form

REVIEWER COMMENTS

Reviewer #1 (Remarks to the Author):

Authors have been constructively incorporated suggestions but provided at places articulation that doesn't reflect the actual comments

For example

MIC by default is calculated and expressed per specific strain unless is otherwise noted (variability per strain and laboratory that tests individual strains is well documented), therefore the use of "specific stain" is excessive unless the authors were presenting collective data and the strains were grouped in different fashion.

The comments regarding the use of KPC and the differences between Gram-positive and negative strains are interconnected. KPC is not another Gram-negative strain but a member of the ESCAPE panel that traditionally is used for new antimicrobial testing. As the authors have an intrinsic deficiency in wet lab study design is not surprising that they can't deduce consistent observations for the susceptibility profile of the newly discovered peptides. Their observations are valid, consistent and improved but the same consistency should be anticipating in explaining data from physical experimentation

Reviewer #2 (Remarks to the Author):

The revision is great to me. Thank you!

Revision 3

Reviewer #1 (Remarks to the Author):

Authors have been constructively incorporated suggestions but provided at places articulation that doesn't reflect the actual comments

For example

MIC by default is calculated and expressed per specific strain unless is otherwise noted (variability per strain and laboratory that tests individual strains is well documented), therefore the use of "specific stain" is excessive unless the authors were presenting collective data and the strains were grouped in different fashion.

The comments regarding the use of KPC and the differences between Gram-positive and negative strains are interconnected. KPC is not another Gram-negative strain but a member of the ESCAPE panel that traditionally is used for new antimicrobial testing. As the authors have an intrinsic deficiency in wet lab study design is not surprising that they can't deduce consistent observations for the susceptibility profile of the newly discovered peptides. Their observations are valid, consistent and improved but the same consistency should be anticipating in explaining data from physical experimentation

Thank you for your valuable comments. We agree that the statement "MIC of a peptide is always measured against a specific strain of bacteria, and it can be different against different bacteria and different strains" may be excessive, as it may be obvious for readers with microbiological background. However, for readers with computational background, this fact may be unknown. Our initial intention when mentioning the "specific strain" was to make sure that the readers know that the MIC values which the model was trained on were determined for *E.coli*. We changed that sentence to "The MIC value for a peptide depends on the peptide's sequence and bacterial strain. ". This sentence occurs in the "HydrAMP --- a conditional, generative model of peptide sequences" section when discussing the training data.

As our model generates new AMPs and is trained on MICs against *E. coli*, the question that arises is – do our newly generated AMPs show different selectivity between different strains (*E. coli* vs other Gram-negative strains or Gram-negative vs Gram-positive strains)?

In order to test this, we have decided to use: MRSA, which is a Gram positive, clinically relevant strain, part of ESKAPE group, and two Gram negative strains (also from the ESKAPE group) – *A. baumannii* and *P. aeruginosa*.

Additional tests on KPC undoubtedly would enrich our study. We agree that this strain and all ESKAPE strains need attention in terms of development of new antibiotics.

However, KPC would be the fifth Gram – negative strain in the experiment. Moreover, there are in general a lot of other Gram-negative or positive strains that could be tested. The added value of further increasing the number of tested strains on top of what was already included is questionable.

In particular, without KPC, our conclusions, derived from experiments that are independent of the fact whether we included KPC or not, are still reliable and valid. Importantly the selected strains are clinically relevant, and with four Gram-negative strains included, they are sufficient to decide whether there are differences between Gram-negative strains or not and whether there are differences between Gram-positives and negatives or not. KPC would not help to give a general statement whether there is a difference between Gram-negatives or positives or not.

On top of that, the addition of KPC would not give us more “consistent observations” because our results are already consistent (see Table 2). We showed that the newly generated peptides have similar activity against two different *E. coli* strains, as expected. We observed an interesting trend that all tested peptides tend to be more active against *A. baumannii* than *P. aeruginosa*. We did not see a clear tendency for the peptides to be more or less active against Gram-positive *S. aureus* versus the remaining, Gram-negative bacterial strains. As already pointed out, this result was expected and was consistent with the literature, as there are examples of peptides which show higher activity against *S. aureus* strains than *P. aeruginosa* (e.g., Omiganan; Fritsche T.R. et al (<https://academic.oup.com/jac/article/61/5/1092/848354>), as well as examples of peptides that show higher activity against *A. baumannii* than against *S. aureus* (e.g., Pexiganan; <https://www.ncbi.nlm.nih.gov/pmc/articles/PMC89207/>).

Therefore we have decided not to extend the biological assay at this stage of the study. Thanks to your comments we will consider usage of KPC in further studies.

Reviewer #2 (Remarks to the Author):

The revision is great to me. Thank you!